# Contextual Multinomial Logit Bandits
# with General Value Functions

**Mengxiao Zhang**
University of Iowa
mengxiao-zhang@uiowa.edu

**Haipeng Luo**
University of Southern California
haipengl@usc.edu

## Abstract

Contextual multinomial logit (MNL) bandits capture many real-world assortment recommendation problems such as online retailing/advertising. However, prior work has only considered (generalized) linear value functions, which greatly limits its applicability. Motivated by this fact, in this work, we consider contextual MNL bandits with a general value function class that contains the ground truth, borrowing ideas from a recent trend of studies on contextual bandits. Specifically, we consider both the stochastic and the adversarial settings, and propose a suite of algorithms, each with different computation-regret trade-off. When applied to the linear case, our results not only are the first ones with no dependence on a certain problem-dependent constant that can be exponentially large, but also enjoy other advantages such as computational efficiency, dimension-free regret bounds, or the ability to handle completely adversarial contexts and rewards.

## 1 Introduction

As assortment recommendation becomes ubiquitous in real-world applications such as online retailing and advertising, the multinomial (MNL) bandit model has attracted great interest in the past decade since it was proposed by Rusmevichientong et al. [24]. It involves a learner and a customer interacting for $T$ rounds. At each round, knowing the reward/profit for each of the $N$ available items, the learner selects a subset/assortment of size at most $K$ and recommend it to the customer, who then purchases one of these $K$ items or none of them according to a multinomial logit model specified by the customer's valuation over the items. The goal of the learner is to learn these unknown valuations over time and select the assortments with high reward.

To better capture practical applications where there is rich contextual information about the items and customers, a sequence of recent works study a contextual MNL bandit model where the customer's valuation is determined by the context via an unknown (generalized) linear function [8, 21, 7, 19, 20, 23, 2]. However, there are no studies on general value functions, despite many recent breakthroughs for classic contextual multi-armed bandits using a general value function class with much stronger representation power that enables fruitful results in both theory and practice [1, 10, 27, 11, 25].

**Contributions.**   Motivated by this gap, we propose a contextual MNL bandit model with a general value function class that contains the ground truth (a standard realizability assumption), and develop a suite of algorithms for different settings and with different computation-regret trade-off.

More specifically, in Section 3, we first consider a stochastic setting where the context-reward pairs are i.i.d. samples of an unknown distribution. Following the work by Simchi-Levi and Xu [25] for contextual bandits, we reduce the problem to an easier offline log loss regression problem and propose two strategies using an offline regression oracle: one with simple and efficient uniform exploration, and another with more adaptive exploration (and hence improved regret) induced by a novel log-barrier regularized strategy. Our results rely on several new technical findings, including a fast

38th Conference on Neural Information Processing Systems (NeurIPS 2024).

Table 1: Comparisons of results for contextual MNL bandits with $T$ rounds, $N$ items, size-$K$ assortments, and a $d$-dimensional linear value function class with norm bounded by $B$. All previous results depend on a problem-dependent constant $\kappa$ that is $\exp(2B)$ in the worst case, while ours (in gray) do not. The notation $\widetilde{\mathcal{O}}(\cdot)$ hides logarithmic dependency on all parameters. In the last column, $\checkmark$ means polynomial runtime in all parameters; $\checkmark\!\!\!\!/$ means polynomial only when $K$ is a constant; and $\boldsymbol{X}$ means not polynomial even for a small $K$.

| Context $x_t$ & reward $r_t$ | Regret | Efficient? |
|---|---|---|
| Stochastic $(x_t, r_t)$ | $\widetilde{\mathcal{O}}((dBNK)^{1/3}T^{2/3})$ (Corollary 3.5) | $\checkmark$ |
| | $\widetilde{\mathcal{O}}(K^2\sqrt{dBNT})$ (Corollary 3.8) | $\checkmark\!\!\!/$ |
| Adversarial $x_t$, $r_t \equiv \mathbf{1}$ | $\widetilde{\mathcal{O}}(dK\sqrt{T/\kappa} + d^2K^4\kappa)$ [23] | $\boldsymbol{X}$ |
| Stochastic $x_t$ Adversarial $r_t$ | $\widetilde{\mathcal{O}}(d\sqrt{T} + d^2K^2\kappa^4)$ [7] | $\checkmark\!\!\!/$ |
| | $\widetilde{\mathcal{O}}(\kappa\sqrt{dT} + \kappa^4)$ [20] | $\boldsymbol{X}$ |
| | $\widetilde{\mathcal{O}}(d\sqrt{\kappa T} + \kappa^2)$ [20] | $\checkmark$ |
| | $\Omega(\max\{\sqrt{dT}, d\sqrt{T}/K\})$ [7] | N/A |
| Adversarial $(x_t, r_t)$ | $\mathcal{O}((NKB)^{1/3}T^{5/6})$ (Corollary 4.4) | $\checkmark$ |
| | $\mathcal{O}(K^2\sqrt{NB}T^{3/4})$ (Corollary 4.7) | $\checkmark\!\!\!/$ |
| | $\widetilde{\mathcal{O}}(K^2\sqrt{dNT})$ (Corollary 4.8) | $\boldsymbol{X}$ |

rate regression result (Lemma 3.1), a "reverse Lipschitzness" for the MNL model (Lemma 3.3), and a certain "low-regret-high-dispersion" property of the log-barrier regularized strategy (Lemma 3.6).

Next, in Section 4, we switch to the more challenging adversarial setting where the context-reward pairs can be arbitrarily chosen. We start by following the idea of [10, 11] for contextual bandits and reducing our problem to online log loss regression, and show that it suffices to find a strategy with a small Decision-Estimation Coefficient (DEC) [10, 14]. We then show that, somewhat surprisingly, the same log-barrier regularized strategy we developed for the stochastic setting leads to a small DEC, despite the fact that it is not the exact DEC minimizer (unlike its counterpart for contextual bandits [13]). We prove this by using the same aforementioned low-regret-high-dispersion property, which to our knowledge is a new way to bound DEC and reveals why log-barrier regularized strategies work in different settings and for different problems. Finally, we also extend the idea of Feel-Good Thompson Sampling [30] and propose a variant for our problem that leads to the best regret bounds in some cases, despite its lack of computational efficiency.

Throughout the paper, we use two running examples to illustrate the concrete regret bounds our different algorithms achieve: the finite class and the linear class. In particular, for the linear class, this leads to five new results, summarized in Table 1 together with previous results. These results all have their own advantages and disadvantages, but we highlight the following:

- While all previous regret bounds depend on a problem-dependent constant $\kappa$ that can be exponentially large in the norm of the weight vector $B$, *none of our results depends on $\kappa$*. In fact, our best results (Corollary 4.8) even has only logarithmic dependence on $B$, a potential *doubly-exponential improvement* compared to prior works.[1]

- The regret bounds of our two algorithms that make use of an online regression oracle are *dimension-free*, despite not having the optimal $\sqrt{T}$-dependence (Corollary 4.4 and Corollary 4.7).

- Our results are the first to handle completely adversarial context-reward pairs.[2]

_______________

[1]One caveat is that, following [5, 9, 18], we assume that no-purchase is the most likely outcome by normalizing the range of the values to $[0, 1]$, making it only a subclass of the one considered in [19, 7, 20]. However, we emphasize that the bounds presented in Table 1 have been translated accordingly to fit our setting. Also note that the $\kappa$ dependence in [23] is in the form of $\sqrt{T/\kappa} + \kappa$ (so not necessarily increasing in $\kappa$), but it only holds for uniform rewards.

[2]Agrawal et al. [2] also considered adversarial contexts and rewards, but there is a technical issue in their analysis as pointed out by the authors. Even if corrected, their results still depend on $\kappa$ while ours do not.

**Related works.** The (non-contextual) MNL model was initially studied in [24], followed by a line of improvements [3, 4, 6, 5, 22]. Specifically, Agrawal et al. [3, 5] introduced a UCB-type algorithm achieving $\widetilde{\mathcal{O}}(\sqrt{NT})$ regret and proved a lower bound of $\Omega(\sqrt{NT/K})$. Subsequently, Chen and Wang [6] enhanced the lower bound to $\Omega(\sqrt{NT})$, matching the upper bound up to log factors.

Cheung and Simchi-Levi [8] first extended MNL bandits to its contextual version and designed a Thompson sampling based algorithm. Follow-up works consider this problem under different settings, including stochastic context [7, 19, 20], adversarial context [21, 2], and uniform reward over items [23]. However, as mentioned, all these works consider (generalized) linear value functions, and our work is the first to consider contextual MNL bandits under a general value function class.

Our work is also closely related to the recent trend of designing contextual bandits algorithms for a general function class. Due to space limit, we defer the discussion to Appendix A.

## 2 Notations and Preliminary

**Notations.** Throughout this paper, we denote the set $\{1, 2, \ldots, N\}$ for some positive integer $N$ by $[N]$ and $\{0, 1, 2, \ldots, N\}$ by $[N]_0$. For a vector $u \in \mathbb{R}^N$, we use $u_i$ to denote its $i$-th coordinate, and for a matrix $W \in \mathbb{R}^{N \times M}$, we use $W_j$ to denote its $j$-th column. For a set $\mathcal{S}$, we denote by $\Delta(\mathcal{S})$ the set of distributions over $\mathcal{S}$, and by $\mathrm{conv}(\mathcal{S})$ the convex hull of $\mathcal{S}$. Finally, for a distribution $\mu \in \Delta([N]_0)$ and an outcome $i \in [N]_0$, the corresponding log loss is $\ell_{\log}(\mu, i) = -\log \mu_i$.

We consider the following contextual MNL bandit problem that proceeds for $T$ rounds. At each round $t$, the learner receives a context $x_t \in \mathcal{X}$ for some arbitrary context space $\mathcal{X}$ and a reward vector $r_t \in [0, 1]^N$ which specifies the reward of $N$ items. Then, out of these $N$ items, the learner needs to recommend a subset $S_t \subseteq \mathcal{S}$ to a customer, where $\mathcal{S} \subseteq 2^{[N]}$ is the collection of all subsets of $[N]$ with cardinality at least 1 and at most $K$ for some $K \leq N$. Finally, the learner observes the customer purchase decision $i_t \in S_t \cup \{0\}$, where 0 denotes the no-purchase option, and receives reward $r_{t,i_t}$, where for notational convenience we define $r_{t,0} = 0$ for all $t$ (no reward if no purchase). The customer decision $i_t$ is assumed to follow an MNL model:

$$\Pr[i_t = i \mid S_t, x_t] = \begin{cases} \frac{f_i^\star(x_t)}{1 + \sum_{j \in S_t} f_j^\star(x_t)} & \text{if } i \in S_t, \\ \frac{1}{1 + \sum_{j \in S_t} f_j^\star(x_t)} & \text{if } i = 0, \\ 0 & \text{otherwise}, \end{cases} \tag{1}$$

where $f^\star : \mathcal{X} \to [0, 1]^N$ is an unknown value function, specifying the costumer's value for each item under the given context. The MNL model above implicitly assumes a value of 1 for the no-purchase option, making it the most likely outcome. This is a standard assumption that holds in many realistic settings [5, 9, 18].

To simplify notation, we define $\mu : \mathcal{S} \times [0, 1]^N \to \Delta([N]_0)$ such that $\mu_i(S, v) \propto v_i \mathbf{1}[i \in S \cup \{0\}]$ with the convention $v_0 = 1$. The purchase decision $i_t$ is thus sampled from the distribution $\mu(S_t, f^\star(x_t))$. In addition, given a reward vector $r \in [0, 1]^N$ (again, with convention $r_0 = 0$), we further define the expected reward of choosing subset $S \in \mathcal{S}$ under context $x \in \mathcal{X}$ as

$$R(S, v, r) = \mathbb{E}_{i \sim \mu(S, v)}[r_i] = \sum_{i \in S} \mu_i(S, v) r_i = \frac{\sum_{i \in S} r_i v_i}{1 + \sum_{i \in S} v_i}.$$

The goal of the learner is then to minimize her regret, defined as the expected gap between her total reward and that of the optimal strategy with the knowledge of $f^\star$:

$$\mathbf{Reg}_{\mathsf{MNL}} = \mathbb{E}\left[\sum_{t=1}^T \max_{S \in \mathcal{S}} R(S, f^\star(x_t), r_t) - \sum_{t=1}^T R(S_t, f^\star(x_t), r_t)\right].$$

To ensure that no-regret is possible, we make the following assumption, which is standard in the literature of contextual bandits.

**Assumption 1** *The learner is given a function class $\mathcal{F} = \{f : \mathcal{X} \to [0, 1]^N\}$ which contains $f^\star$.*

Our hope is thus to design algorithms whose regret is sublinear in $T$ and polynomial in $N$ and some standard complexity measure of the function class $\mathcal{F}$. So far, we have not specified how the

**Algorithm 1** Contextual MNL Algorithms with an Offline Regression Oracle

---

Input: an offline regression oracle $\mathsf{Alg}_{\mathsf{off}}$ satisfying Assumption 2
Define: epoch schedule $\tau_0 = 0$ and $\tau_m = 2^{m-1} - 1$ for all $m = 1, 2, \ldots$.
**for** *epoch* $m = 1, 2, \ldots$ **do**
  Feed $\{x_t, S_t, i_t\}_{t=\tau_{m-1}+1}^{\tau_m}$ to $\mathsf{Alg}_{\mathsf{off}}$ and obtain $f_m$.
  Define a stochastic policy $q_m : \mathcal{X} \times [0,1]^N \to \Delta(\mathcal{S})$ via either Eq. (4) or Eq. (5).
  **for** $t = \tau_m + 1, \cdots, \tau_{m+1}$ **do**
    Observe context $x_t \in \mathcal{X}$ and reward vector $r_t \in [0,1]^N$.
    Sample $S_t \sim q_m(x_t, r_t)$ and recommend it to the customer.
    Observe customer's purchase decision $i_t \in S_t \cup \{0\}$, drawn according to Eq. (1).

---

context $x_t$ and the reward $x_t$ are chosen. In the next two sections, we will discuss both the easier stochastic case where $(x_t, r_t)$ is jointly drawn from some fixed and unknown distribution, and the harder adversarial case where $(x_t, r_t)$ can be arbitrarily chosen by an adversary.

## 3   Contextual MNL Bandits with Stochastic Contexts and Rewards

In this section, we consider contextual MNL bandits with stochastic contexts and rewards, where at each round $t \in [T]$, $x_t$ and $r_t$ are jointly drawn from a fixed and unknown distribution $\mathcal{D}$. Following the literature of contextual bandits, we aim to reduce the problem to an easier and better-studied offline regression problem and only access the function class $\mathcal{F}$ through some offline regression oracle. Specifically, an offline regression oracle $\mathsf{Alg}_{\mathsf{off}}$ takes as input a set of i.i.d. context-subset-purchase tuples and outputs a predictor from $\mathcal{F}$ with low generalization error in terms of log loss, formally defined as follows.

**Assumption 2** *Given $n$ samples $D = \{(x_k, S_k, i_k)\}_{k=1}^n$ where each $(x_k, S_k, i_k) \in \mathcal{X} \times \mathcal{S} \times [N]_0$ is an i.i.d. sample of some unknown distribution $\mathcal{H}$ and the conditional distribution of $i_k$ is $\mu(S_k, f^\star(x_k))$, with probability at least $1 - \delta$ the offline regression oracle $\mathsf{Alg}_{\mathsf{off}}$ outputs a function $\widehat{f}_D \in \mathcal{F}$ such that:*

$$\mathbb{E}_{(x,S,i) \sim \mathcal{H}} \left[ \ell_{\log}(\mu(S, \widehat{f}_D(x)), i) - \ell_{\log}(\mu(S, f^\star(x)), i) \right] \le \mathbf{Err}_{\log}(n, \delta, \mathcal{F}), \tag{2}$$

*for some function $\mathbf{Err}_{\log}(n, \delta, \mathcal{F})$ that is non-increasing in $n$.*

Given the similarity between MNL and multi-class logistic regression, assuming such a log loss regression oracle is more than natural. Indeed, in the following lemma, we prove that for both the finite class and a certain linear function class, the empirical risk minimizer (ERM) not only satisfies this assumption, but also enjoys a fast $1/n$ rate. The proof is based on the observation that our loss function $\ell_{\log}(\mu(S, f(x)), i)$, when seen as a function of $f$, satisfies the so-called strong 1-central condition [17, Definition 7], which might be of independent interest; see Appendix B.1 for details.

**Lemma 3.1** *The ERM strategy $\widehat{f}_D = \operatorname{argmin}_{f \in \mathcal{F}} \sum_{(x,S,i) \in D} \ell_{\log}(\mu(S, f(x)), i)$ satisfies Assumption 2 for the following two cases:*

- *(Finite class) $\mathcal{F}$ is a finite class of functions with image $[\beta, 1]^N$ for some $\beta \in (0, 1)$ and $\mathbf{Err}_{\log}(n, \delta, \mathcal{F}) = \mathcal{O}\left( \frac{\log K/\beta \log |\mathcal{F}|/\delta}{n} \right)$.*

- *(Linear class) $\mathcal{X} \subseteq \{x \in \mathbb{R}^{d \times N} \mid \|x_i\|_2 \le 1, \ \forall i \in [N]\}$, $\mathcal{F} = \{f_{\theta,i}(x) = e^{\theta^\top x_i - B} \mid \|\theta\|_2 \le B\}$, and $\mathbf{Err}_{\log}(n, \delta, \mathcal{F}) = \mathcal{O}\left( \frac{dB \log K \log(Bn) \log \frac{1}{\delta}}{n} \right)$, for some $B > 0$.[3]*

---

[3]We call this a linear class (even though it is technically log-linear) because, when combined with the MNL model Eq. (1), it becomes the standard softmax model with linear policies. Also note that the bias term $-B$ in the exponent makes sure $f_\theta(x) \in [0,1]^N$. Following [23], we assume $\|\theta\|_2 \le B$ instead of $\|\theta\|_2 \le 1$ to ensure the representation power of the function class, since we already normalize the contexts and restrict them to be within the unit ball.

Due to space limit, we only use these two simple function classes as running examples throughout the paper, but we emphasize that our results can be applied to any class as long as regression is feasible. For additional examples, see Appendix D.

Given $\mathsf{Alg}_{\mathsf{off}}$, we now outline a natural algorithm framework that proceeds in epochs with exponentially increasing length (see Algorithm 1): At the beginning of each epoch $m$, the algorithm feeds all the context-subset-purchase tuples from the last epoch to the offline regression oracle $\mathsf{Alg}_{\mathsf{off}}$ and obtains a value predictor $f_m$. Then, it decides in some way using $f_m$ a stochastic policy $q_m$, which maps a context $x$ and a reward vector $r \in [0,1]^N$ to a distribution over $\mathcal{S}$. With such a policy in hand, for every round $t$ within this epoch, the algorithm simply samples a subset $S_t$ according to $q_m(x_t, r_t)$ and recommend it to the customer.

We will specify two concrete stochastic policies $q_m$ in the next two subsections. Before doing so, we highlight some key parts of the analysis that shed light on how to design a "good" $q_m$. The first step is an adaptation of Simchi-Levi and Xu [25, Lemma 7], which quantifies the expected reward difference of any policy under the ground-truth value function $f^\star$ versus the estimated value function $f_m$. Specifically, for a deterministic policy $\pi : \mathcal{X} \times [0,1]^N \to \mathcal{S}$ mapping from a context-reward pair to a subset, we define its true expected reward and its expected reward under $f_m$ respectively as (overloading the notation $R$):

$$R(\pi) = \mathbb{E}_{(x,r)\sim\mathcal{D}}\left[R(\pi(x,r), f^\star(x), r)\right], \quad R_m(\pi) = \mathbb{E}_{(x,r)\sim\mathcal{D}}\left[R(\pi(x,r), f_m(x), r)\right]. \quad (3)$$

Moreover, for any $\rho \in \Delta(\mathcal{S})$, define $w(\rho) \in [0,1]^N$ such that $w_i(\rho) = \sum_{S\in\mathcal{S}:i\in S}\rho(S)$ is the probability of item $i$ being selected under distribution $\rho$, and for any stochastic policy $q$, further define a dispersion measure for a deterministic policy $\pi$ as $V(q,\pi) = \mathbb{E}_{(x,r)\sim\mathcal{D}}\left[\sum_{i\in\pi(x,r)}\frac{1}{w_i(q(x,r))}\right]$ (the smaller $V(q,\pi)$ is, the more disperse the distribution induced by $q$ is). Using the Lipschitzness (in $v$) of the reward function $R(S,v,r)$ (Lemma B.1), we prove the following.

**Lemma 3.2** *For any deterministic policy $\pi : \mathcal{X} \times [0,1]^N \to \mathcal{S}$ and any epoch $m \geq 2$, we have*

$$|R_m(\pi) - R(\pi)| \leq \sqrt{V(q_{m-1},\pi)} \cdot \sqrt{\mathbb{E}_{(x,r)\sim\mathcal{D},S\sim q_{m-1}(x,r)}\left[\sum_{i\in S}\left(f_{m,i}(x) - f_i^\star(x)\right)^2\right]}.$$

If the learner could observe the true value of each item in the selected subset (or its noisy version), then doing squared loss regression on these values would make the squared loss term in Lemma 3.2 small; this is essentially the case in the contextual bandit problem studied by Simchi-Levi and Xu [25]. However, in our problem, only the purchase decisions are observed but not the true values that define the MNL model. Nevertheless, one of our key technical contributions is to show that the offline log-loss regression, which only relies on observing the purchase decisions, in fact also makes sure that the squared loss above is small.

**Lemma 3.3** *For any $S \in \mathcal{S}$ and $v, v^\star \in [0,1]^N$, we have*

$$\frac{1}{2(K+1)^4}\sum_{i\in S}(v_i - v_i^\star)^2 \leq \|\mu(S,v) - \mu(S,v^\star)\|_2^2 \leq 2\mathbb{E}_{i\sim\mu(S,v^\star)}\left[\ell_{\log}(\mu(S,v),i) - \ell_{\log}(\mu(S,v^\star),i)\right].$$

The first equality establishes certain "reverse Lipschitzness" of $\mu$ and is proven by providing a universal lower bound on the minimum singular value of its Jacobian matrix, which is new to our knowledge. It implies that if two value vectors induce a pair of close distributions, then they must be reasonably close as well. The second equality, proven using known facts, further states that to control the distance between two distributions, it suffices to control their log loss difference, which is exactly the job of the offline regression oracle.

Therefore, combining Lemma 3.2 and Lemma 3.3, we see that to design a good algorithm, it suffices to find a stochastic policy that "mostly" follows $\mathrm{argmax}_S R(S, f_m(x_t), r_t)$, the best decision according to the oracle's prediction, and at the same time ensures high dispersion for all $\pi$ such that the oracle's predicted reward for any policy is close to its true reward. The design of our two algorithms in the remaining of this section follows exactly this principle.

## 3.1 A Simple and Efficient Algorithm via Uniform Exploration

As a warm-up, we first introduce a simple but efficient $\varepsilon$-greedy-type algorithm that ensures reasonable dispersion by uniformly exploring all the singleton sets. Specifically, at epoch $m$, given the value predictor $f_m$ from $\mathsf{Alg}_{\mathsf{off}}$, $q_m(x,r) \in \Delta(\mathcal{S})$ is defined as follows for some $\varepsilon_m > 0$:

$$q_m(S|x,r) = (1-\varepsilon_m)\mathbb{1}\left[S = \operatorname*{argmax}_{S^\star \in \mathcal{S}} R(S^\star, f_m(x), r)\right] + \frac{\varepsilon_m}{N}\sum_{i=1}^N \mathbb{1}\left[S = \{i\}\right]. \qquad (4)$$

In other words, with probability $1 - \varepsilon$, the learner picks the subset achieving the maximum reward based on the reward vector $r$ and the predicted value $f_m(x)$; with the remaining $\varepsilon$ probability, the learner selects a uniformly random item $i \in [N]$ and recommend only this item, which clearly ensures $V(q_m, \pi) \le \frac{KN}{\varepsilon_m}$ for any $\pi$. Based on our previous analysis, it is straightforward to prove the following regret guarantee.

**Theorem 3.4** *Under Assumption 1 and Assumption 2, Algorithm 1 with $q_m$ defined in Eq. (4) and the optimal choice of $\varepsilon_m$ ensures $\mathbf{Reg}_{\mathsf{MNL}} = \sum_{m=1}^{\lceil \log_2 T\rceil}\mathcal{O}\left(2^m(NK\mathbf{Err}_{\log}(2^{m-1}, 1/T^2, \mathcal{F}))^{\frac{1}{3}}\right)$.*

To better interpret this regret bound, we consider the finite class and the linear class discussed in Lemma 3.1. Combining it with Theorem 3.4, we immediately obtain the following corollary:

**Corollary 3.5** *Under Assumption 1, Algorithm 1 with $q_m$ defined in Eq. (4), the optimal choice of $\varepsilon_m$, and ERM as $\mathsf{Alg}_{\mathsf{off}}$ ensures $\mathbf{Reg}_{\mathsf{MNL}} = \mathcal{O}\left((NK\log\frac{K}{\beta}\log(|\mathcal{F}|T))^{\frac{1}{3}}T^{\frac{2}{3}}\right)$ for finite class and $\mathbf{Reg}_{\mathsf{MNL}} = \mathcal{O}\left((dBNK\log K)^{\frac{1}{3}}T^{\frac{2}{3}}\log(BT)\log T\right)$ for linear class (see Lemma 3.1 for definitions).*

While these $\widetilde{\mathcal{O}}(T^{2/3})$ regret bounds are suboptimal, Theorem 3.4 provides the first computationally efficient algorithms for contextual MNL bandits with an offline regression oracle for a general function class. Indeed, computing $\operatorname{argmax}_{S^\star \in \mathcal{S}} R(S^\star, f_m(x), r)$ can be efficiently done in $\mathcal{O}(N^2)$ time according to [24, Section 2.1]. Moreover, for the linear case, the ERM oracle can indeed be efficiently (and approximately) implemented because it is a convex optimization problem over a simple ball constraint. Importantly, previous regret bounds for the linear case all depend on a problem-dependent constant $\kappa = \max_{\|\theta\|\le B, S\in\mathcal{S}, i\in S, t\in[T]}\frac{1}{\mu_i(S, f_\theta(x_t))\mu_0(S, f_\theta(x_t))}$, which is $\exp(2B)$ in the worst case [7, 20, 23], but ours only has polynomial dependence on $B$.

## 3.2 Better Exploration Leads to Better Regret

Next, we show that a more sophisticated construction of $q_m$ in Algorithm 1 leads to better exploration and consequently improved regret bounds. Specifically, $q_m$ is defined as (for some $\gamma_m > 0$):

$$q_m(x,r) = \operatorname*{argmax}_{\rho \in \Delta(\mathcal{S})}\mathbb{E}_{S\sim\rho}\left[R(S, f_m(x), r)\right] - \frac{(K+1)^4}{\gamma_m}\sum_{i=1}^N \log\frac{1}{w_i(\rho)}. \qquad (5)$$

The first term of the optimization objective above is the expected reward when one picks a subset according to $\rho$ and the value function is $f_m$, while the second term is a certain log-barrier regularizer applied to $\rho$, penalizing it for putting too little mass on any single item. This specific form of regularization ensures that $q_m$ enjoys a low-regret-high-dispersion guarantee, as shown below.

**Lemma 3.6** *For any $x \in \mathcal{X}$ and $r \in [0,1]^N$, the distribution $q_m(x,r)$ defined in Eq. (5) satisfies:*

$$\max_{S^\star \in \mathcal{S}} R(S^\star, f_m(x), r) - \mathbb{E}_{S\sim q_m(x,r)}\left[R(S, f_m(x), r)\right] \le \frac{N(K+1)^4}{\gamma_m}, \qquad (6)$$

$$\forall S \in \mathcal{S}, \quad \sum_{i\in S}\frac{1}{w_i(q_m(x,r))} \le N + \frac{\gamma_m}{(K+1)^4}\left(\max_{S^\star \in \mathcal{S}} R(S^\star, f_m(x), r) - R(S, f_m(x), r)\right). \qquad (7)$$

Eq. (6) states that following $q_m(x,r)$ does not incur too much regret compared to the best subset predicted by the oracle, and Eq. (7) states that the dispersion of $q_m(x,r)$ on any subset is controlled

by how bad this subset is compared to the best one in terms of their predicted reward — a good subset has a large dispersion while a bad one can have a smaller dispersion since we do not care about estimating its true reward very accurately. Such a refined dispersion guarantee intuitively provides a much more adaptive exploration scheme compared to uniform exploration.

This kind of low-regret-high-dispersion guarantees is in fact very similar to the ideas of Simchi-Levi and Xu [25] for contextual bandits (which itself is similar to an earlier work by Agarwal et al. [1]). While Simchi-Levi and Xu [25] were able to provide a closed-form strategy with such a guarantee for contextual bandits, we do not find a similar closed-form for MNL bandits and instead provide the strategy as the solution of an optimization problem Eq. (5). Unfortunately, we are not aware of an efficient way to solve Eq. (5) with polynomial time complexity, but one can clearly solve it in $\text{poly}(|\mathcal{S}|) = \text{poly}(N^K)$ time since it is a concave problem over $\Delta(\mathcal{S})$. Thus, the algorithm is efficient when $K$ is small, which we believe is the case for most real-world applications.

Combining Lemma 3.2 and Lemma 3.6, we prove the following regret guarantee, which improves the $\mathbf{Err}_{\log}^{1/3}$ term in Theorem 3.4 to $\mathbf{Err}_{\log}^{1/2}$ (proofs deferred to Appendix B).

**Theorem 3.7** *Under Assumption 1 and Assumption 2, Algorithm 1 with $q_m$ defined in Eq. (5) and the optimal choice of $\gamma_m$ ensures $\mathbf{Reg}_{\mathsf{MNL}} = \mathcal{O}\left(\sum_{m=1}^{\lceil \log_2 T \rceil} 2^m K^2 \sqrt{N \mathbf{Err}_{\log}(2^{m-1}, 1/T^2, \mathcal{F})}\right)$.*

Similar to Section 3.1, we instantiate Theorem 3.7 using the following two concrete classes:

**Corollary 3.8** *Under Assumption 1, Algorithm 1 with $q_m$ defined in Eq. (5), the optimal choice of $\gamma_m$, and ERM as $\mathsf{Alg}_{\mathsf{off}}$ ensures $\mathbf{Reg}_{\mathsf{MNL}} = \mathcal{O}\left(K^2 \sqrt{T \log \frac{K}{\beta} \log(|\mathcal{F}|T)}\right)$ for the finite class and $\mathbf{Reg}_{\mathsf{MNL}} = \mathcal{O}\left(K^2 \sqrt{dBNT \log(BT) \log T}\right)$ for the linear class (see Lemma 3.1 for definitions).*

The dependence on $T$ in these $\mathcal{O}(\sqrt{T})$ regret bounds is known to be optimal [6, 7]. Once again, in the linear case, we have no exponential dependence on $B$, unlike previous results.

## 4 Contextual MNL Bandits with Adversarial Contexts and Rewards

In this section, we move on to consider the more challenging case where the context $x_t$ and the reward vector $r_t$ can both be arbitrarily chosen by an adversary. We propose two different approaches leading to three different algorithms, each with its own pros and cons.

### 4.1 First Approach: Reduction to Online Regression

In the first approach, we follow a recent trend of studies that reduces contextual bandits to online regression and only accesses $\mathcal{F}$ through an online regression oracle [10, 11, 15, 31, 29]. More specifically, we assume access to an online regression oracle $\mathsf{Alg}_{\mathsf{on}}$ that follows the protocol below: at each round $t \in [T]$, $\mathsf{Alg}_{\mathsf{on}}$ outputs a value predictor $f_t \in \text{conv}(\mathcal{F})$; then, it receives a context $x_t$, a subset $S_t$, and a purchase decision $i_t \in S_t \cup \{0\}$, all chosen arbitrarily, and suffers log loss $\ell_{\log}(\mu(S_t, f_t(x_t)), i_t)$.[4] The oracle is assumed to enjoy the following regret guarantee.

**Assumption 3** *The predictions made by the online regression oracle $\mathsf{Alg}_{\mathsf{on}}$ ensure:*

$$\mathbb{E}\left[\sum_{t=1}^{T} \ell_{\log}(\mu(S_t, f_t(x_t)), i_t) - \sum_{t=1}^{T} \ell_{\log}(\mu(S_t, f^\star(x_t)), i_t)\right] \leq \mathbf{Reg}_{\log}(T, \mathcal{F}),$$

*for any $f^\star \in \mathcal{F}$ and some regret bound $\mathbf{Reg}_{\log}(T, \mathcal{F})$ that is non-decreasing in $T$.*

While most previous works on contextual bandits assume a squared loss online oracle, log loss is more than natural for our MNL model (it was also used by Foster and Krishnamurthy [11] to achieve first-order regret guarantees for contextual bandits). The following lemma shows that Assumption 3 again holds for the finite class and the linear class.

---

[4]In fact, for our purpose, $i_t$ is always sampled from $\mu(S_t, f^\star(x_t))$, instead of being chosen arbitrarily, but the concrete oracle examples we provide in Lemma 4.1 indeed work for arbitrary $i_t$.

**Lemma 4.1** *For the finite class and the linear class discussed in Lemma 3.1, the following concrete oracles satisfy Assumption 3:*

- *(Finite class) Hedge [16] with* $\mathbf{Reg}_{\log}(T, \mathcal{F}) = \mathcal{O}(\sqrt{T \log |\mathcal{F}|} \log \frac{K}{\beta})$;

- *(Linear class) Online Gradient Descent [32] with* $\mathbf{Reg}_{\log}(T, \mathcal{F}) = \mathcal{O}(B\sqrt{T})$.

Unfortunately, unlike the offline oracle, we are not able to provide a "fast rate" (that is, $\widetilde{\mathcal{O}}(1)$ regret) for these two cases, because our loss function does not appear to satisfy the standard Vovk's mixability condition or any other sufficient conditions discussed in Van Erven et al. [26]. This is in sharp contrast to the standard multi-class logistic loss [12], despite the similarity between these two models. We leave as an open problem whether fast rates exist for these two classes, which would have immediate consequences to our final MNL regret bounds below.

With this online regression oracle, a natural algorithm framework works as follows: at each round $t$, the learner first obtains a value predictor $f_t \in \mathrm{conv}(\mathcal{F})$ from the regression oracle $\mathsf{Alg}_{\mathsf{on}}$; then, upon seeing context $x_t$ and reward vector $r_t$, the learner decides in some way a distribution $q_t \in \Delta(\mathcal{S})$ based on $f_t(x_t)$ and $r_t$, and samples $S_t$ from $q_t$; finally, the learner observes the purchase decision $i_t$ and feeds the tuple $(x_t, S_t, i_t)$ to the oracle $\mathsf{Alg}_{\mathsf{on}}$ (see Algorithm 2 in Appendix C). To shed light on how to design a good sampling distribution $q_t$, we show a general lemma that holds for any $q_t$.

**Lemma 4.2** *Under Assumption 1 and Assumption 3, Algorithm 2 (with any $q_t$) ensures*

$$\mathbf{Reg}_{\mathsf{MNL}} \leq \mathbb{E}\left[\sum_{t=1}^{T} \mathsf{dec}_\gamma(q_t; f_t(x_t), r_t)\right] + 2\gamma \mathbf{Reg}_{\log}(T, \mathcal{F})$$

*for any $\gamma > 0$, where $\mathsf{dec}_\gamma(q; v, r)$ is the Decision-Estimation Coefficient (DEC) defined as*

$$\max_{v^\star \in [0,1]^N} \max_{S^\star \in \mathcal{S}} \left\{ R(S^\star, v^\star, r) - \mathbb{E}_{S \sim q}\left[R(S, v^\star, r)\right] - \gamma \mathbb{E}_{S \sim q}\left[\|\mu(S, v) - \mu(S, v^\star)\|_2^2\right] \right\}. \quad (8)$$

Our DEC adopts the idea of Foster et al. [14] for general decision making problems: the term $R(S^\star, v^\star, r) - \mathbb{E}_{S \sim q}\left[R(S, v^\star, r)\right]$ represents the instantaneous regret of strategy $q$ against the best subset $S^\star$ with respect to reward vector $r$ and the worst-case value vector $v^\star$, and the term $\mathbb{E}_{S \sim q}[\|\mu(S, v) - \mu(S, v^\star)\|_2^2]$ is the expected squared distance between two distributions induced by $v$ and $v^\star$, which, in light of the second inequality of Lemma 3.3, lower bounds the instantaneous log loss regret of the online oracle. Therefore, a small DEC makes sure that the learner's MNL regret is somewhat close to the oracle's log loss regret $\mathbf{Reg}_{\log}$, formally quantified by Lemma 4.2. With the goal of ensuring a small DEC, we again propose two strategies similar to Section 3.

**Uniform Exploration.** We start with a simple uniform exploration approach similar to Eq. (4):

$$q_t(S) = (1 - \varepsilon)\mathbb{1}\left[S = \operatorname*{argmax}_{S^\star \in \mathcal{S}} R(S^\star, f_t(x_t), r_t)\right] + \frac{\varepsilon}{N}\sum_{i=1}^{N}\mathbb{1}\left[S = \{i\}\right]. \quad (9)$$

where $\varepsilon > 0$ is a parameter specifying the probability of uniformly exploring the singleton sets. We prove the following results for this simple algorithm.

**Theorem 4.3** *The strategy defined in Eq. (9) guarantees* $\mathsf{dec}_\gamma(q_t; f_t(x_t), r_t) = \mathcal{O}(\frac{NK}{\gamma \varepsilon} + \varepsilon)$. *Consequently, under Assumption 1 and Assumption 3, Algorithm 2 with $q_t$ calculated via Eq. (9) and the optimal choice of $\varepsilon$ and $\gamma$ ensures* $\mathbf{Reg}_{\mathsf{MNL}} = \mathcal{O}\left((NK\mathbf{Reg}_{\log}(T, \mathcal{F}))^{\frac{1}{3}}T^{\frac{2}{3}}\right)$.

Combining this with Lemma 4.1, we immediately obtain the following corollary.

**Corollary 4.4** *Under Assumption 1, Algorithm 2 with $q_t$ defined in Eq. (9) and the optimal choice of $\varepsilon$ and $\gamma$ ensures* $\mathbf{Reg}_{\mathsf{MNL}} = \mathcal{O}\left((NK \log \frac{K}{\beta})^{\frac{1}{3}}T^{\frac{5}{6}}\right)$ *for the finite class (with Hedge as $\mathsf{Alg}_{\mathsf{on}}$) and* $\mathbf{Reg}_{\mathsf{MNL}} = \mathcal{O}\left((NKB)^{\frac{1}{3}}T^{\frac{5}{6}}\right)$ *for the linear class (with Online Gradient Descent as $\mathsf{Alg}_{\mathsf{on}}$).*

While these regret bounds have a large dependence on $T$, the advantage of this algorithm is its computational efficiency as discussed before.

**Better Exploration.** Can we improve the algorithm via a strategy with an even smaller DEC? In particular, what happens if we take the extreme and let $q_t$ be the minimizer of $\text{dec}_\gamma(q; f_t(x_t), r_t)$? Indeed, this is exactly the approach in several prior works that adopt the DEC framework [13, 29], where the exact minimizer for DEC is characterized and shown to achieve a small DEC value.

On the other hand, for our problem, it appears quite difficult to analyze the exact DEC minimizer. Somewhat surprisingly, however, we show that the same construction in Eq. (5) for the stochastic environment in fact also achieves a reasonably small DEC for the adversarial case:

**Theorem 4.5** *The following distribution satisfies* $\text{dec}_\gamma(q_t, f_t(x_t), r_t) \leq \mathcal{O}\left(\frac{NK^4}{\gamma}\right)$:

$$q_t = \underset{q \in \Delta(\mathcal{S})}{\operatorname{argmax}} \, \mathbb{E}_{S \sim q}\left[R(S, f_t(x_t), r_t)\right] - \frac{(K+1)^4}{\gamma} \sum_{i=1}^{N} \log \frac{1}{w_i(q)}. \tag{10}$$

A couple of remarks are in order. First, while for some cases such as the contextual bandit problem studied by Foster et al. [13], this kind of log-barrier regularized strategies is known to be the exact DEC minimizer, one can verify that this is not the case for our DEC. Second, the fact that the same strategy works for both the stochastic and the adversarial environments is similar to the case for contextual bandits where the same inverse gap weighting strategy works for both cases [10, 25], but to our knowledge, the connection between these two cases is unclear since their analysis is quite different. Finally, our proof (in Appendix C) in fact relies on the same low-regret-high-dispersion property of Lemma 3.6, which is a new way to bound DEC as far as we know. More importantly, this to some extent demystifies the last two points: the reason that such log-barrier regularized strategies work regardless whether they are the exact minimizer or not and regardless whether the environment is stochastic or adversarial is all due to their inherent low-regret-high-dispersion property.

Combining Theorem 4.5 with Lemma 4.2, we obtain the following improved regret.

**Theorem 4.6** *Under Assumption 1 and Assumption 3, Algorithm 2 with $q_t$ calculated via Eq. (10) and the optimal choice of $\gamma$ ensures* $\textbf{Reg}_{\text{MNL}} = \mathcal{O}\left(K^2 \sqrt{NT\textbf{Reg}_{\log}(T, \mathcal{F})}\right)$.

**Corollary 4.7** *Under Assumption 1, Algorithm 2 with $q_t$ defined in Eq. (10) and the optimal choice of $\gamma$ ensures* $\textbf{Reg}_{\text{MNL}} = \mathcal{O}\left(K^2 \sqrt{N \log \frac{K}{\beta}} T^{\frac{3}{4}} (\log|\mathcal{F}|)^{\frac{1}{4}}\right)$ *for the finite class (with Hedge as* $\text{Alg}_{\text{on}}$*) and* $\textbf{Reg}_{\text{MNL}} = \mathcal{O}\left(K^2 \sqrt{NB} T^{\frac{3}{4}}\right)$ *for the linear class (with Online Gradient Descent as* $\text{Alg}_{\text{on}}$*).*

We remark that if the "fast rate" discussed after Lemma 4.1 exists, we would have obtained the optimal $\sqrt{T}$-regret here. Despite having worse dependence on $T$, however, our result for the linear case enjoys three advantages compared to prior work [7, 20, 23]: 1) no exponential dependence on $B$ (as in all our other results), 2) no dependence at all on the dimension $d$, and 3) valid even when contexts and rewards are adversarial. We refer the reader to Table 1 again for detailed comparisons.

## 4.2 Second Approach: Feel-Good Thompson Sampling

The second approach we take is to extend the idea of the Feel-Good Thompson Sampling algorithm of Zhang [30] for contextual bandits. Due to space limit, we defer the algorithm and its analysis to Appendix E, and only state its regret bounds for the finite class and the linear class (a corollary of a more general regret bound in Theorem E.1).

**Corollary 4.8** *Under Assumption 1, Algorithm 3 wensures* $\textbf{Reg}_{\text{MNL}} = \mathcal{O}\left(K^2 \sqrt{NT \log|\mathcal{F}|}\right)$ *for the finite class and* $\textbf{Reg}_{\text{MNL}} = \mathcal{O}\left(K^2 \sqrt{dNT \log(BTK)}\right)$ *for the linear class.*

In terms of the dependence on $T$, Algorithm 3 achieves the best (and in fact optimal) regret bounds among all our results. For the linear case, it even has only logarithmic dependence on $B$, a potential doubly-exponential improvement compared to prior works. The caveat is that there is no efficient way to implement the algorithm even for the linear case and even when $K$ is a constant (unlike all our other algorithms). We leave the question of whether there exists a computationally efficient algorithm (even only for small $K$) with a $\sqrt{T}$-regret bound that has no exponential dependence on $B$ as a key future direction.

# 5 Conclusion and Future Directions

In this work, we consider contextual MNL bandits with a general value function class under a realizability assumption. For both the stochastic and the adversarial settings, we propose a suite of algorithms with different computational-regret trade-off. Notably, none of our regret bounds suffers from the exponentially large dependence on some problem dependent constant in the case with linear value functions. One interesting future direction is to improve the $\mathrm{poly}(K, N)$ dependence in our regret upper bounds, which seems to require new techniques.

## Acknowledgments and Disclosure of Funding

The authors were supported by NSF Award IIS-1943607.

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

# A Additional Related Works

As mentioned, our work is closely related to the recent trend of designing contextual bandits algorithms for a general function class. Specifically, under stochastic context, Xu and Zeevi [27], Simchi-Levi and Xu [25] designed algorithms based on an offline squared loss regression oracle and achieved optimal regret guarantees. Under adversarial context, there are two lines of works. The first one reduces the contextual bandit problem to online regression [10, 11, 14, 31, 29], while the second one is based on the ability to sample from a certain distribution over the function class using Markov chain Monte Carlo methods [30, 28]. We follow and greatly extend the ideas of all these approaches to design algorithms for contextual MNL bandits.

# B Omitted Details in Section 3

## B.1 Offline Regression Oracle

We start by proving Lemma 3.1, which shows that ERM strategy satisfies Assumption 2 for the finite class and the linear function class.

**Proof** [of Lemma 3.1] We first show that our log loss function $\ell_{\log}(\mu(S, f(x)), i)$ satisfies the so-called strong 1-central condition (Definition 7 of Grünwald and Mehta [17]), which states that there exists $f_0 \in \mathcal{F}$, such that for any $f \in \mathcal{F}$,

$$\mathbb{E}_{(x,S,i)\sim\mathcal{H}}\left[\exp(-(\ell_{\log}(\mu(S, f(x)), i) - \ell_{\log}(\mu(S, f_0(x)), i)))\right] \leq 1.$$

Indeed, by picking $f_0 = f^\star$, we know that

$$\mathbb{E}_{(x,S,i)\sim\mathcal{H}}\left[\exp(-(\ell_{\log}(\mu(S, f(x), i)) - \ell_{\log}(S, f^\star(x), i)))\right]$$

$$= \mathbb{E}_{(x,S)}\mathbb{E}_{i\sim\mu(S,f^\star(x))}\left[\frac{\mu_i(S, f)}{\mu_i(S, f^\star)}\right]$$

$$= \mathbb{E}_{(x,S)}\left[\sum_{i\in S\cup\{0\}}\mu_i(S, f)\right] = 1,$$

certifying the strong 1-central condition.

Now, we first consider the case where $\mathcal{F}$ is finite. Since $f_i(x) \geq \beta$ for all $x \in \mathcal{X}$ and $i \in [N]$, we know that for any $i \in [N]_0$, we have (defining $f_0(x) = 1$)

$$\ell_{\log}(\mu(S, f(x)), i) = \log\frac{1 + \sum_{j\in S} f_j(x)}{f_i(x)} \leq \log\frac{K + 1}{\beta}.$$

Therefore, according to Theorem 7.6 of [26], we know that given $n$ i.i.d samples $D = \{(x_k, S_k, i_k)\}_{k\in[n]}$, ERM predictor $\widehat{f}_D$ guarantees that with probability $1 - \delta$:

$$\mathbb{E}_{(x,S,i)\sim\mathcal{D}}\left[\ell_{\log}(\mu(S, \widehat{f}_D(x)), i)\right] \leq \mathbb{E}_{(x,S,i)\sim\mathcal{D}}\left[\ell_{\log}(\mu(S, f^\star(x)), i)\right] + \mathcal{O}\left(\frac{\log\frac{K}{\beta}\log\frac{|\mathcal{F}|}{\delta}}{n}\right).$$

Next, we consider the linear function class. In this case, we know that $x_i^\top\theta - B \in [-2B, 0]$ for all $x_i$. Therefore, $\ell_{\log}(\mu(S, f(x)), i)$ is bounded by $2B + 2\ln N$ for all $x \in \mathcal{X}, f \in \mathcal{F}, S \in \mathcal{S}$ and $i \in [N]$ since

$$\ell_{\log}(\mu(S, f(x)), i) = \log\frac{1 + \sum_{j\in S}\exp(x_j^\top\theta - B)}{\exp(x_i^\top\theta - B)} \leq \log\frac{1 + K}{e^{-2B}} \leq 2B + 2\log K,$$

and the same bound clearly holds as well for $i = 0$. Moreover, since

$$\left\|\nabla_\theta\log\frac{1 + \sum_{j\in S}\exp(x_j^\top\theta - B)}{\exp(x_i^\top\theta - B)}\right\|_2 = \left\|\frac{\sum_{j\in S}\exp(\theta^\top x_j - B)x_j}{1 + \sum_{j\in S}\exp(\theta^\top x_j - B)} - x_i\right\|_2 \leq 2,$$

we know that the $\varepsilon$-covering number of $\ell_{\log} \circ \mathcal{F} \triangleq \{\ell_{\log}^f : f \in \mathcal{F}\}$ is bounded by $\left(\frac{16B}{\varepsilon}\right)^d$, where with an abuse of notation, we define $Z \triangleq (x, S, i)$ and denote $\ell_{\log}(\mu(S, f(x)), i)$ by $\ell_{\log}^f(Z)$. Therefore, according to Theorem 7.7 of [26], we know that given $n$ i.i.d samples $D = \{(x_k, S_k, i_k)\}_{k \in [n]}$, ERM predictor $\widehat{f}_D$ guarantees that with probability $1 - \delta$:

$$\mathbb{E}_{(x,S,i)\sim\mathcal{D}} \left[ \ell_{\log}(\mu(S, \widehat{f}_D(x)), i) \right] \leq \mathbb{E}_{(x,S,i)\sim\mathcal{D}} \left[ \ell_{\log}(\mu(S, f^\star(x)), i) \right] + \mathcal{O}\left(\frac{dB \log K \log(Bn) \log \frac{1}{\delta}}{n}\right).$$

$\square$

### B.2 Analysis of Algorithm 1

We first prove the following lemma, which shows that the expected reward function $R(S, v, r)$ is 1-Lipschitz in the value vector $v$.

**Lemma B.1** *Given $r \in [0, 1]^N$ and $S \subseteq [N]$, function $R(S, v, r) = \frac{\sum_{i \in S} r_i v_i}{1 + \sum_{i \in S} v_i}$ satisfies that for any $v', v \in [0, \infty)^N$, $|R(S, v, r) - R(S, v', r)| \leq \sum_{i \in S} |v_i - v_i'|$.*

**Proof** Taking derivative with respect to $v_j$ for $j \in S$, we know that

$$\left| \nabla_{v_j} R(S, v, r) \right| = \left| \frac{r_j(1 + \sum_{i \in S} v_i) - \sum_{j \in S} r_j v_j}{(1 + \sum_{i \in S} v_i)^2} \right| \leq \max\left\{ \frac{r_j}{1 + \sum_{i \in S} v_i}, \frac{\sum_{i \in S} v_i}{(1 + \sum_{i \in S} v_i)^2} \right\} \leq 1,$$

where both inequalities are because $r_j \in [0, 1]$. This finishes the proof. $\square$

Next, we restate and prove Lemma 3.2.

**Lemma B.2** *For any deterministic policy $\pi : \mathcal{X} \times [0, 1]^N \to \mathcal{S}$ and any epoch $m \geq 2$, we have*

$$|R_m(\pi) - R(\pi)| \leq \sqrt{V(q_{m-1}, \pi)} \cdot \sqrt{\mathbb{E}_{(x,r)\sim\mathcal{D}, S \sim q_{m-1}(x,r)} \left[ \sum_{i \in S} (f_{m,i}(x) - f_i^\star(x))^2 \right]}.$$

**Proof** We proceed as:

$$|R_m(\pi) - R(\pi)|$$
$$= \left| \mathbb{E}_{(x,r)\sim\mathcal{D}} \left[ R(\pi(x,r), f_m(x), r) - R(\pi(x,r), f^\star(x), r) \right] \right|$$
$$\leq \mathbb{E}_{(x,r)\sim\mathcal{D}} \left[ \sum_{i=1}^{N} \mathbb{1}\{i \in \pi(x,r)\} |f_{m,i}(x) - f_i^\star(x)| \right] \qquad (11)$$
$$\leq \mathbb{E}_{(x,r)\sim\mathcal{D}} \left[ \sqrt{\sum_{i=1}^{N} \frac{\mathbb{1}\{i \in \pi(x,r)\}}{w_i(q_{m-1}|x,r)} \sum_{i=1}^{N} w_i(q_{m-1}|x,r) \left( f_{m,i}(x) - f_i^\star(x) \right)^2} \right]$$
$$\text{(CauchySchwarz inequality)}$$
$$\leq \sqrt{\mathbb{E}_{(x,r)\sim\mathcal{D}} \left[ \sum_{i=1}^{N} \frac{\mathbb{1}\{i \in \pi(x,r)\}}{w_i(q_{m-1}|x,r)} \right]} \cdot \sqrt{\mathbb{E}_{(x,r)\sim\mathcal{D}} \left[ \sum_{i=1}^{N} w_i(q_{m-1}|x,r) \left( f_{m,i}(x) - f_i^\star(x) \right)^2 \right]}$$
$$\text{(CauchySchwarz inequality)}$$
$$= \sqrt{V(q_{m-1}, \pi)} \cdot \sqrt{\mathbb{E}_{(x,r)\sim\mathcal{D}} \left[ \sum_{i=1}^{N} w_i(q_{m-1}|x,r) \left( f_{m,i}(x) - f_i^\star(x) \right)^2 \right]}$$
$$= \sqrt{V(q_{m-1}, \pi)} \cdot \sqrt{\mathbb{E}_{(x,r)\sim\mathcal{D}, S \sim q_{m-1}(x,r)} \left[ \sum_{i \in S} (f_{m,i}(x) - f_i^\star(x))^2 \right]}, \qquad (12)$$

where the first inequality uses the convexity of the absolute value function and Lemma B.1. $\square$

Next, to prove Lemma 3.3, we first prove the following key technical lemma (where $\mathbf{1}$ denotes the all-one vector).

**Lemma B.3** *Let* $h(a) = \frac{a}{1+\mathbf{1}^\top a}$ *for* $a \in [0,1]^d$. *Then, for any* $a, b \in [0,1]^d$, *we have*

$$\frac{1}{2(d+1)^4}\|a-b\|_2^2 \leq \|h(a) - h(b)\|_2^2.$$

**Proof** The Jacobian matrix of $h$ is

$$H(a) = \frac{1}{1+\mathbf{1}^\top a}\mathbf{I} - \frac{\mathbf{1}a^\top}{(1+\mathbf{1}^\top a)^2}.$$

Therefore, there exists $z \in \text{conv}(\{a,b\})$ such that $\|h(a) - h(b)\|_2 = \|H(z)(a-b)\|_2$. It thus remains to figure out the minimum singular value of $H(z)$, which is equal to the reciprocal of the spectral norm of $H(z)^{-1}$. By Sherman-Morrison formula, we know that

$$H(z)^{-1} = (1+\mathbf{1}^\top z)(\mathbf{I} + \mathbf{1}z^\top).$$

Therefore, we have

$$\begin{aligned}
H(z)^{-1}H(z)^{-\top} &= (1+\mathbf{1}^\top z)^2(\mathbf{I}+\mathbf{1}z^\top)(\mathbf{I}+\mathbf{1}z^\top)^\top \\
&= (1+\mathbf{1}^\top z)^2(\mathbf{I}+\mathbf{1}z^\top + z\mathbf{1}^\top + z^\top z\mathbf{1}\mathbf{1}^\top).
\end{aligned}$$

Note that for any $u$ that is perpendicular to the subspace spanned by $\{z, \mathbf{1}\}$, we have $H(z)^{-1}H(z)^{-\top}u = (1+\mathbf{1}^\top z)^2 u$. Therefore, there are $d-2$ identical eigenvalues 1 for the matrix $\frac{1}{(1+\mathbf{1}^\top z)^2}H(z)^{-1}H(z)^{-\top}$. Let the remaining two eigenvalues of $\frac{1}{(1+\mathbf{1}^\top z)^2}H(z)^{-1}H(z)^{-\top}$ be $\lambda_1$ and $\lambda_2$. Note that

$$\begin{aligned}
\lambda_1\lambda_2 &= \det\left((\mathbf{I}+\mathbf{1}z^\top)(\mathbf{I}+z\mathbf{1}^\top)\right) = (1+\mathbf{1}^\top z)^2, \\
\lambda_1 + \lambda_2 &= \text{Trace}(\mathbf{I}+\mathbf{1}z^\top + z\mathbf{1}^\top + z^\top z\mathbf{1}\mathbf{1}^\top) - (d-2) \\
&= 2 + 2\mathbf{1}^\top z + z^\top z\mathbf{1}^\top\mathbf{1} \\
&= 2 + 2\mathbf{1}^\top z + d\cdot z^\top z \\
&\leq 2 + 2d + d^2.
\end{aligned}$$

Therefore, we know that $\max\{\lambda_1, \lambda_2\} \leq \lambda_1 + \lambda_2 \leq 2 + 2d + d^2$, meaning that

$$\|H(z)^{-1}H(z)^{-\top}\|_2 \leq 2(1+\mathbf{1}^\top z)^2(1+d+d^2) \leq 2(1+d)^2(d^2+d+1) \leq 2(d+1)^4.$$

This further means that the minimum singular value of $H(z)$ is at least $\frac{1}{\sqrt{2}(d+1)^2}$. Therefore, we can conclude that

$$\|h(a) - h(b)\|_2 \geq \frac{1}{\sqrt{2}(d+1)^2}\|a-b\|_2,$$

leading to

$$\|h(a) - h(b)\|_2^2 \geq \frac{1}{2(d+1)^4}\|a-b\|_2^2.$$

□ Next, we restate and prove Lemma 3.3.

**Lemma B.4** *For any* $S \in \mathcal{S}$ *and* $v, v^\star \in [0,1]^N$, *we have*

$$\frac{1}{2(K+1)^4}\sum_{i\in S}(v_i - v_i^\star)^2 \leq \|\mu(S,v) - \mu(S,v^\star)\|_2^2 \leq 2\mathbb{E}_{i\sim\mu(S,v^\star)}\left[\ell_{\log}(\mu(S,v),i) - \ell_{\log}(\mu(S,v^\star),i)\right].$$

**Proof** The first inequality follows directly from Lemma B.3 using the fact that $|S| \leq K$ for all $S \in \mathcal{S}$. Consider the second inequality. For any $\mu, \mu' \in \Delta([K])$, by definition of $\ell_{\log}(\mu,i)$, we know that

$$\mathbb{E}_{i\sim\mu}\left[\ell_{\log}(\mu',i) - \ell_{\log}(\mu,i)\right] = \mathbb{E}_{i\sim\mu}\left[\log\frac{\mu_i}{\mu_i'}\right] = \text{KL}(\mu,\mu') \geq \frac{1}{2}\|\mu-\mu'\|_1^2 \geq \frac{1}{2}\|\mu-\mu'\|_2^2,$$

where the first inequality is due to Pinsker's inequality. □

### B.3 Omitted Details in Section 3.1

In this section, we show omitted details in Section 3.1. For ease of presentation, we assume that the distribution over context-reward pair $\mathcal{D}$ has finite support. All our results can be directly generalized to the case with infinite support following a similar argument in Appendix A.7 of [25]. Define $\Psi : \mathcal{X} \times [0,1]^N \mapsto \mathcal{S}$ as the set of all deterministic policy. Following Lemma 3 in [25], we know that for any context $x \in \mathcal{X}$ and reward vector $r \in [0,1]^N$, and any stochastic policy $q : \mathcal{X} \times [0,1]^N \mapsto \Delta(\mathcal{S})$, there exists an equivalent randomized policy $Q \in \Delta(\Psi)$ such that for all $S \in \mathcal{S}$, $x \in \mathcal{X}$, and $r \in [0,1]^N$,

$$q(S|x,r) = \sum_{\pi \in \Psi} \mathbb{1}\{\pi(x,r) = S\}Q(\pi).$$

Let $Q_m$ be the randomized policy induced by $q_m$. Define $\text{Reg}(\pi)$ and $\text{Reg}_m(\pi)$ as:

$$\text{Reg}(\pi) = R(\pi_{f^\star}) - R(\pi), \quad \text{Reg}_m(\pi) = R_m(\pi_{f_m}) - R_m(\pi), \tag{13}$$

where $R(\pi)$ and $R_m(\pi)$ are defined in Eq. (3) and $\pi_f$ is the policy that maps each $(x,r)$ to the one-hot distribution supported on $\text{argmax}_{S \in \mathcal{S}} R(S, f(x), r)$.

Following the analysis in [25], we show that to analyze our algorithms expected regret, we only need to analyze the induced randomized policies implicit regret.

**Lemma B.5** *Fix any epoch $m$. For any round $t$ in this epoch, we have*

$$\mathbb{E}_{(x_t,r_t) \sim \mathcal{D}, S_t \sim q_m(x_t,r_t)} \left[ R(\pi_{f^\star}(x_t,r_t), f^\star(x), r_t) - R(S_t, f^\star(x), r_t) \right] = \sum_{\pi \in \Psi} Q_m(\pi)\text{Reg}(\pi).$$

**Proof** Direct calculation shows that

$$\mathbb{E}_{(x_t,r_t) \sim \mathcal{D}, S_t \sim q_m(x_t,r_t)} \left[ R(\pi_{f^\star}(x_t,r_t), f^\star(x), r_t) - R(S_t, f^\star(x), r_t) \right]$$

$$= \mathbb{E}_{(x_t,r_t) \sim \mathcal{D}} \left[ R(\pi_{f^\star}(x_t,r_t), f^\star(x), r_t) - \sum_{S \in \mathcal{S}} q_m(S|x_t,r_t)R(S, f^\star(x), r_t) \right]$$

$$= \mathbb{E}_{(x_t,r_t) \sim \mathcal{D}} \left[ R(\pi_{f^\star}(x_t,r_t), f^\star(x), r_t) - \sum_{S \in \mathcal{S}} \sum_{\pi \in \Psi} \mathbb{1}\{\pi(x_t,r_t) = S\}Q_m(\pi)R(S, f^\star(x), r_t) \right]$$

$$= \mathbb{E}_{(x,r) \sim \mathcal{D}} \left[ \sum_{S \in \mathcal{S}} \sum_{\pi \in \Psi} \mathbb{1}\{\pi(x,r) = S\}Q_m(\pi) \left( R(\pi_{f^\star}(x,r), f^\star(x), r) - R(S, f^\star(x), r) \right) \right]$$

$$= \mathbb{E}_{(x,r) \sim \mathcal{D}} \left[ \sum_{\pi \in \Psi} Q_m(\pi) \left( R(\pi_{f^\star}(x,r), f^\star(x), r) - R(\pi(x,r), f^\star(x), r) \right) \right]$$

$$= \sum_{\pi \in \Psi} Q_m(\pi)\mathbb{E}_{(x,r) \sim \mathcal{D}} \left[ R(\pi_{f^\star}(x,r), f^\star(x), r) - R(\pi(x,r), f^\star(x), r) \right]$$

$$= \sum_{\pi \in \Psi} Q_m(\pi)\text{Reg}(\pi),$$

which finishes the proof. □

To prove our main results for Algorithm 1, we define the following good event:

**Event 1** *For all epoch $m \geq 2$, $f_m$ satisfies*

$$\mathbb{E}_{(x,r) \sim \mathcal{D}, S \sim q_{m-1}(x,r), i \sim \mu(S, f^\star(x))} \left[ \ell_{\log}(\mu(S, f_m(x)), i) - \ell_{\log}(\mu(S, f^\star(x)), i) \right]$$

$$\leq \mathbf{Err}_{\log}(\tau_m - \tau_{m-1}, 1/T^2, \mathcal{F}).$$

According to Assumption 2, Event 1 happens with probability at least $1 - \frac{1}{T}$ since there are at most $T$ epochs.

Although now we have all ingredients to analyze our $\varepsilon$-greedy-type algorithm defined Eq. (4), to get the exact result in Theorem 3.4, we will in fact need a refined version of Lemma 3.2, which eventually provides a tighter regret guarantee.

**Lemma B.6** *Suppose that Event 1 holds. Algorithm 1 with $q_t$ defined in Eq. (4) satisfies that for any deterministic policy $\pi \in \Psi$ and any epoch $m \geq 2$, we have*

$$|R_m(\pi) - R(\pi)| \leq 8\sqrt{\frac{NK}{\varepsilon_{m-1}}} \cdot \sqrt{\mathbf{Err}_{\log}(2^{m-2}, 1/T^2, \mathcal{F})}.$$

**Proof** Following Eq. (11) in the proof of Lemma 3.2, we know that

$$|R_m(\pi) - R(\pi)|$$

$$\leq \mathbb{E}_{(x,r)\sim\mathcal{D}}\left[\sum_{i=1}^{N} \mathbb{1}\{i \in \pi(x,r)\}|f_{m,i}(x) - f_i^\star(x)|\right]$$

$$\leq \mathbb{E}_{(x,r)\sim\mathcal{D}}\left[\sqrt{\sum_{i=1}^{N} \frac{N\mathbb{1}\{i \in \pi(x,r)\}}{\varepsilon_{m-1}} \sum_{i=1}^{N} \frac{\varepsilon_{m-1}}{N}(f_{m,i}(x) - f_i^\star(x))^2}\right]$$

(CauchySchwarz inequality)

$$\leq \sqrt{\mathbb{E}_{(x,r)\sim\mathcal{D}}\left[\sum_{i=1}^{N} \frac{N\mathbb{1}\{i \in \pi(x,r)\}}{\varepsilon_{m-1}}\right]} \cdot \sqrt{\mathbb{E}_{(x,r)\sim\mathcal{D}}\left[\sum_{i=1}^{N} \frac{\varepsilon_{m-1}}{N}(f_{m,i}(x) - f_i^\star(x))^2\right]}$$

(CauchySchwarz inequality)

$$\leq \sqrt{\frac{NK}{\varepsilon_{m-1}}} \cdot \sqrt{\mathbb{E}_{(x,r)\sim\mathcal{D}}\left[\sum_{i=1}^{N} \frac{\varepsilon_{m-1}}{N}(f_{m,i}(x) - f_i^\star(x))^2\right]}. \tag{14}$$

Since $f_m$ is the output of $\mathsf{Alg}_{\mathsf{off}}$ with i.i.d tuples $\{(x_t, S_t, i_t)\}_{t=\tau_{m-1}+1}^{\tau_m}$, according to Lemma 3.3 and Event 1, we know that

$$64\mathbf{Err}_{\log}(\tau_m - \tau_{m-1}, 1/T^2, \mathcal{F})$$

$$\geq 32\mathbb{E}_{(x,r)\sim\mathcal{D}, S\sim q_{m-1}(x,r)}\left[\|\mu(S, f_m(x)) - \mu(S, f^\star(x))\|_2^2\right] \qquad \text{(Lemma 3.3)}$$

$$\geq \frac{32\varepsilon_{m-1}}{N} \sum_{i=1}^{N} \mathbb{E}_{(x,r)\sim\mathcal{D}}\left[\|\mu(\{i\}, f_m(x)) - \mu(\{i\}, f^\star(x))\|_2^2\right] \qquad \text{(according to Eq. (4))}$$

$$\geq \frac{\varepsilon_{m-1}}{N} \sum_{i=1}^{N} \mathbb{E}_{(x,r)\sim\mathcal{D}}\left[\sum_{i=1}^{N}(f_{m,i}(x) - f_i^\star(x))^2\right]. \qquad \text{(using Lemma B.3 with } d = 1)$$

Plugging the above inequality back to Eq. (14) and noticing that $\tau_m = 2^{m-1} - 1$, we know that

$$|R_m(\pi) - R(\pi)| \leq 8\sqrt{\frac{NK}{\varepsilon_{m-1}} \cdot \mathbf{Err}_{\log}(2^{m-2}, 1/T^2, \mathcal{F})}.$$

$\square$

Now we are ready to prove Theorem 3.4

**Theorem 3.4** *Under Assumption 1 and Assumption 2, Algorithm 1 with $q_m$ defined in Eq. (4) and the optimal choice of $\varepsilon_m$ ensures $\mathbf{Reg}_{\mathsf{MNL}} = \sum_{m=1}^{\lceil \log_2 T \rceil} \mathcal{O}\left(2^m(NK\mathbf{Err}_{\log}(2^{m-1}, 1/T^2, \mathcal{F}))^{\frac{1}{3}}\right)$.*

**Proof** Consider the regret within epoch $m \geq 2$. Under Event 1, we know that for any $\pi \in \Psi$,

$$\mathrm{Reg}(\pi) = R(\pi_{f^\star}) - R(\pi)$$

$$= (R(\pi_{f^\star}) - R_m(\pi_{f_m})) - (R_m(\pi) - R_m(\pi_{f_m})) + (R_m(\pi) - R(\pi))$$

$$\leq (R(\pi_{f^\star}) - R_m(\pi_{f^\star})) + (R_m(\pi_{f_m}) - R_m(\pi)) + (R_m(\pi) - R(\pi))$$

$$\leq (R_m(\pi_{f_m}) - R_m(\pi)) + 16\sqrt{\frac{NK}{\varepsilon_{m-1}} \cdot \mathbf{Err}_{\log}(2^{m-2}, 1/T^2, \mathcal{F})}, \tag{15}$$

where the first inequality is because $R_m(\pi_{f_m}) \geq R_m(\pi_{f^\star})$ by definition and the second inequality is due to Lemma B.6. Taking summation over all rounds within epoch $m$ and picking $\varepsilon_m = (NK)^{\frac{1}{3}} \mathbf{Err}_{\log}^{\frac{1}{3}}(2^{m-2}, 1/T^2, \mathcal{F})$, we know that

$$
\mathbb{E}\left[ \sum_{t=\tau_m+1}^{\tau_{m+1}} \left( \max_{S \in \mathcal{S}} R(S, x_t, f^\star(x_t)) - R(S_t, x_t, f^\star(x_t)) \right) \right]
$$

$$
= (\tau_{m+1} - \tau_m)\mathbb{E}\left[ \sum_{\pi \in \Psi} Q_m(\pi)\mathrm{Reg}(\pi) \right] \qquad \text{(Lemma B.5)}
$$

$$
\overset{(i)}{\leq} 2^{m-1} \cdot \mathbb{E}\left[ ((1 - \varepsilon_m)\mathrm{Reg}(\pi_{f_m}) + \varepsilon_m) \right]
$$

$$
\overset{(ii)}{\leq} \frac{2^{m-1}}{T} + 2^{m-1}\mathbb{E}\left[ ((1 - \varepsilon_m)\mathrm{Reg}(\pi_{f_m}) + \varepsilon_m) \;\middle|\; \text{Event 1 holds} \right]
$$

$$
\overset{(iii)}{\leq} \frac{2^{m-1}}{T} + 2^{m-1}\left( \varepsilon_m + 16\sqrt{\frac{NK}{\varepsilon_{m-1}} \cdot \mathbf{Err}_{\log}(2^{m-2}, 1/T^2, \mathcal{F})} \right)
$$

$$
\overset{(iv)}{=} \frac{2^{m-1}}{T} + \mathcal{O}\left( 2^{m-1} \left( NK\mathbf{Err}_{\log}(2^{m-2}, 1/T^2, \mathcal{F}) \right)^{\frac{1}{3}} \right),
$$

where $(i)$ is due to $\tau_m = 2^{m-1} - 1$ and the construction of $q_m(x, r)$ defined in Eq. (4); $(ii)$ is because Event 1 holds with probability at least $1 - \frac{1}{T}$; $(iii)$ uses Eq. (15); and $(iv)$ is due to the choice of $\varepsilon_m$. Taking summation over all $m = 2, 3, \ldots \lceil \log_2 T \rceil + 1$ epochs, we can obtain that

$$
\mathbf{Reg}_{\mathsf{MNL}} = \sum_{m=1}^{\lceil \log_2 T \rceil} \mathcal{O}\left( 2^m \left( NK\mathbf{Err}_{\log}(2^{m-1}, 1/T^2, \mathcal{F}) \right)^{\frac{1}{3}} \right).
$$

$\square$

## B.4   Omitted Details in Section 3.2

First, we restate and prove Lemma 3.6, which shows that $q_m$ defined in Eq. (5) enjoys a low-regret-high-dispersion guarantee.

**Lemma B.7**  *For any $x \in \mathcal{X}$ and $r \in [0, 1]^N$, the distribution $q_m(x, r)$ defined in Eq. (5) satisfies:*

$$
\max_{S^\star \in \mathcal{S}} R(S^\star, f_m(x), r) - \mathbb{E}_{S \sim q_m(x,r)}\left[ R(S, f_m(x), r) \right] \leq \frac{N(K+1)^4}{\gamma_m}, \qquad (6)
$$

$$
\forall S \in \mathcal{S}, \quad \sum_{i \in S} \frac{1}{w_i(q_m(x, r))} \leq N + \frac{\gamma_m}{(K+1)^4}\left( \max_{S^\star \in \mathcal{S}} R(S^\star, f_m(x), r) - R(S, f_m(x), r) \right). \quad (7)
$$

**Proof**   It is direct to see that solving Eq. (5) is equivalent to solving the following optimization problem:

$$
\underset{\rho \in \Delta(\mathcal{S})}{\mathrm{argmin}}\, \mathbb{E}_{S \sim \rho}\left[ \max_{S^\star \in \mathcal{S}} R(S^\star, f_m(x), r) - R(S, f_m(x), r) \right] + \frac{(K+1)^4}{\gamma_m} \sum_{i=1}^{N} \log \frac{1}{w_i(\rho)}. \qquad (16)
$$

Moreover, relaxing the constraint $\rho$ from $\Delta(\mathcal{S})$ to $\left\{ \rho \in [0, 1]^{\mathcal{S}} : \sum_{S \in \mathcal{S}} \rho(S) \leq 1 \right\}$ in Eq. (16) does not change the solution, since for any $\rho \in [0, 1]^{\mathcal{S}}$ such that $\sum_{S \in \mathcal{S}} \rho(S) < 1$, putting the remaining $1 - \sum_{S \in \mathcal{S}} \rho(S)$ probability mass on $\mathrm{argmax}_{S^\star \in \mathcal{S}} R(S^\star, f_m(x), r)$ can only make the objective smaller.

Now, consider the Lagrangian form of Eq. (16) over this relaxed constraint and set the derivative with respect to $\rho(S)$ to zero. We obtain

$$
\max_{S^\star \in \mathcal{S}} R(S^\star, f_m(x), r) - R(S, f_m(x), r) - \frac{(K+1)^4}{\gamma_m} \sum_{i:i \in S} \frac{1}{w_i(\rho)} - \lambda(S) + \lambda = 0, \qquad (17)
$$

where $\lambda \geq 0$ and $\lambda(S) \geq 0$, $S \in \mathcal{S}$ are the Lagrangian multipliers. Let $\rho^\star \in \Delta(\mathcal{S})$ be the optimal solution of Eq. (16). Replacing $\rho$ by $\rho^\star$ in Eq. (17), multiplying Eq. (17) by $\rho^\star(S)$ for each $S \in \mathcal{S}$, and taking the summation over $S \in \mathcal{S}$, we know that

$$\sum_{S \in \mathcal{S}} \rho^\star(S) \left( \max_{S^\star \in \mathcal{S}} R(S^\star, f_m(x), r) - R(S, f_m(x), r) \right)$$
$$- \frac{(K+1)^4}{\gamma_m} \sum_{S \in \mathcal{S}} \rho^\star(S) \sum_{i: i \in S} \frac{1}{w_i(\rho^\star)} - \sum_{S \in \mathcal{S}} \rho^\star(S) \lambda(S) + \lambda = 0.$$

Rearranging the terms, we know that

$$\sum_{S \in \mathcal{S}} \rho^\star(S) \left( \max_{S^\star \in \mathcal{S}} R(S^\star, f_m(x), r) - R(S, f_m(x), r) \right)$$
$$= \frac{(K+1)^4}{\gamma_m} \sum_{S \in \mathcal{S}} \rho^\star(S) \sum_{i: i \in S} \frac{1}{w_i(\rho^\star)} + \sum_{S \in \mathcal{S}} \rho^\star(S) \lambda(S) - \lambda$$
$$= \frac{(K+1)^4}{\gamma_m} \sum_{i=1}^{N} \frac{1}{w_i(\rho^\star)} \sum_{S \in \mathcal{S}: i \in S} \rho^\star(S) - \lambda \qquad \text{(complementary slackness)}$$
$$= \frac{N(K+1)^4}{\gamma_m} - \lambda \leq \frac{N(K+1)^4}{\gamma_m},$$

proving Eq. (6). The above also implies that $\lambda \leq \frac{N(K+1)^4}{\gamma_m}$ since

$$\sum_{S \in \mathcal{S}} \rho^\star(S) \left( \max_{S^\star \in \mathcal{S}} R(S^\star, f_m(x), r) - R(S, f_m(x), r) \right) \geq 0.$$

Therefore, Eq. (17) implies that for any $S \in \mathcal{S}$,

$$\sum_{i: i \in S} \frac{1}{w_i(\rho^\star)} = \frac{\gamma_m}{(K+1)^4} \left( \max_{S^\star \in \mathcal{S}} R(S^\star, f_m(x), r) - R(S, f_m(x), r) - \lambda_S + \lambda \right)$$
$$\leq \frac{\gamma_m}{(K+1)^4} \left( \max_{S^\star \in \mathcal{S}} R(S^\star, f_m(x), r) - R(S, f_m(x), r) \right) + N,$$

where the last inequality uses the fact that $\lambda \leq \frac{N(K+1)^4}{\gamma_m}$ and $\lambda_S \geq 0$. This proves Eq. (7). $\qquad \square$
Now, to prove Theorem 3.7, we first prove the following lemma, which shows that the regret with respect to the true value function $f^\star$ and the one respect to the value predictor $f_m$ is within a factor of 2 plus an additional term of order $\frac{N(K+1)^4}{\gamma_m}$.

**Lemma B.8** *Suppose that Event 1 holds. For all epochs $m \geq 2$, all rounds $t$ in this epoch, and all policies $\pi \in \Psi$, with $\gamma_m = \max \left\{ 1, \sqrt{\frac{N(K+1)^4}{\mathbf{Err}_{\log}(2^{m-2}, 1/T^2, \mathcal{F})}} \right\}$ and $\lambda = 33$, we have*

$$\mathrm{Reg}(\pi) \leq 2 \cdot \mathrm{Reg}_m(\pi) + \frac{\lambda N(K+1)^4}{\gamma_m},$$
$$\mathrm{Reg}_m(\pi) \leq 2 \cdot \mathrm{Reg}(\pi) + \frac{\lambda N(K+1)^4}{\gamma_m}.$$

**Proof** We prove this by induction. The base case holds trivially. Suppose that this holds for all epochs with index less than $m$. Consider epoch $m$. We first show that $\mathrm{Reg}(\pi) \leq 2\mathrm{Reg}_m(\pi) + \frac{\lambda N(K+1)^4}{\gamma_m}$ for all deterministic policy $\pi \in \Psi$. This holds trivially if $\sqrt{\frac{N(K+1)^4}{\mathbf{Err}_{\log}(2^{m-2}, 1/T^2, \mathcal{F})}} \leq 1$ since $\mathrm{Reg}(\pi) \leq 1$. Consider the case in which $\gamma_m = \sqrt{\frac{N(K+1)^4}{\mathbf{Err}_{\log}(2^{m-2}, 1/T^2, \mathcal{F})}}$. Specifically, we have

$$\mathrm{Reg}(\pi) - \mathrm{Reg}_m(\pi)$$
$$= (R(\pi_{f^\star}) - R(\pi)) - (R_m(\pi_{f_m}) - R_m(\pi))$$

$$\stackrel{(i)}{\leq} (R(\pi_{f^\star}) - R(\pi)) - (R_m(\pi_{f^\star}) - R_m(\pi))$$

$$\leq |R_m(\pi_{f^\star}) - R(\pi_{f^\star})| + |R_m(\pi) - R(\pi)|$$

$$\stackrel{(ii)}{\leq} \sqrt{V(q_{m-1}, \pi_{f^\star}) \cdot \mathbb{E}_{(x,r)\sim\mathcal{D},\ S\sim q_{m-1}(x,r)} \left[\sum_{i\in S} (f_{m,i}(x) - f_i^\star(x))^2\right]}$$

$$+ \sqrt{V(q_{m-1}, \pi) \cdot \mathbb{E}_{(x,r)\sim\mathcal{D},\ S\sim q_{m-1}(x,r)} \left[\sum_{i\in S} (f_{m,i}(x) - f_i^\star(x))^2\right]}, \tag{18}$$

where $(i)$ is because $R_m(\pi_{f_m}) \geq R_m(\pi_{f^\star})$ by definition and $(ii)$ follows Lemma 3.2. Next, using Lemma 3.3 and Lemma B.3, since Event 1 holds, we know that

$$4(K+1)^4 \mathbf{Err}_{\log}(2^{m-2}, 1/T^2, \mathcal{F})$$

$$\geq 2(K+1)^4 \mathbb{E}_{(x,r)\sim\mathcal{D},\ S\sim q_{m-1}(x,r)} \left[\|\mu(S, f_m(x)) - \mu(S, f^\star(x))\|_2^2\right] \qquad \text{(Lemma 3.3)}$$

$$\geq \mathbb{E}_{(x,r)\sim\mathcal{D},\ S\sim q_{m-1}(x,r)} \left[\sum_{i\in S} (f_{m,i}(x) - f_i^\star(x))^2\right]. \qquad \text{(Lemma B.3)}$$

Plugging the above back to Eq. (18), we obtain that

$$\mathrm{Reg}(\pi) - \mathrm{Reg}_m(\pi) \tag{19}$$

$$\leq 2(K+1)^2 \sqrt{V(q_{m-1}, \pi_{f^\star}) \mathbf{Err}_{\log}(2^{m-2}, 1/T^2, \mathcal{F})}$$

$$\quad + 2(K+1)^2 \sqrt{V(q_{m-1}, \pi) \mathbf{Err}_{\log}(2^{m-2}, 1/T^2, \mathcal{F})}$$

$$\leq \frac{(K+1)^4 V(q_{m-1}, \pi_{f^\star})}{8\gamma_m} + \frac{(K+1)^4 V(q_{m-1}, \pi)}{8\gamma_m} + 16\gamma_m \mathbf{Err}_{\log}(2^{m-2}, 1/T^2, \mathcal{F})$$

$$\text{(AM-GM inequality)}$$

$$= \frac{(K+1)^4 V(q_{m-1}, \pi_{f^\star})}{8\gamma_m} + \frac{(K+1)^4 V(q_{m-1}, \pi)}{8\gamma_m} + \frac{16N(K+1)^4}{\gamma_m}, \tag{20}$$

where the last equality is because $\gamma_m = \sqrt{\frac{N(K+1)^4}{\mathbf{Err}_{\log}(2^{m-2}, 1/T^2, \mathcal{F})}}$. According to Lemma 3.6, we know that for all $\pi \in \Psi$,

$$V(q_{m-1}, \pi) = \mathbb{E}_{(x,r)\sim\mathcal{D}} \left[\sum_{i\in\pi(x,r)} \frac{1}{w_i(q_{m-1}|x,r)}\right]$$

$$\leq \mathbb{E}_{(x,r)\sim\mathcal{D}} \left[N + \frac{\gamma_{m-1}}{(K+1)^4} \left(\max_{S^\star\in\mathcal{S}} R(S^\star, r, f_{m-1}(x)) - R(S, r, f_{m-1}(x))\right)\right]$$

$$= N + \frac{\gamma_{m-1}}{(K+1)^4} \mathrm{Reg}_{m-1}(\pi). \tag{21}$$

Using Eq. (21), we bound the first and the second term in Eq. (20) as follows

$$\frac{(K+1)^4 V(q_{m-1}, \pi)}{8\gamma_m} \leq \frac{N(K+1)^4}{8\gamma_m} + \frac{\gamma_{m-1} \mathrm{Reg}_{m-1}(\pi)}{8\gamma_m}$$

$$\leq \frac{N(K+1)^4}{8\gamma_m} + \frac{\gamma_{m-1} \left(2\mathrm{Reg}(\pi) + \frac{\lambda N(K+1)^4}{\gamma_{m-1}}\right)}{8\gamma_m}$$

$$\leq \frac{1}{4}\mathrm{Reg}(\pi) + \frac{\lambda+1}{8\gamma_m} \cdot N(K+1)^4, \qquad \text{(since } \gamma_{m-1} \leq \gamma_m\text{)}$$

$$\frac{(K+1)^4 V(q_{m-1}, \pi_{f^\star})}{8\gamma_m} \leq \frac{N(K+1)^4}{8\gamma_m} + \frac{\gamma_{m-1} \mathrm{Reg}_{m-1}(\pi_{f^\star})}{8\gamma_m}$$

$$\leq \frac{N(K+1)^4}{8\gamma_m} + \frac{\gamma_{m-1} \left(2\mathrm{Reg}(\pi_{f^\star}) + \frac{\lambda N(K+1)^4}{\gamma_{m-1}}\right)}{8\gamma_m}$$

$$\leq \frac{\lambda+1}{8\gamma_m} \cdot N(K+1)^4. \qquad \text{(since } \mathrm{Reg}(\pi_{f^\star}) = 0 \text{ and } \gamma_{m-1} \leq \gamma_m)$$

Plugging back to Eq. (20), we know that

$$\mathrm{Reg}(\pi) - \mathrm{Reg}_m(\pi) \leq \frac{1}{4}\mathrm{Reg}(\pi) + \frac{16N(K+1)^4}{\gamma_m} + \frac{\lambda+1}{4\gamma_m}N(K+1)^4.$$

Rearranging the terms, we know that

$$\mathrm{Reg}(\pi) \leq \frac{4}{3}\mathrm{Reg}_m(\pi) + \frac{12N(K+1)^4}{\gamma_m} + \frac{\lambda+1}{3\gamma_m}N(K+1)^4$$

$$\leq 2\mathrm{Reg}_m(\pi) + \frac{\lambda N(K+1)^4}{\gamma_m}, \tag{22}$$

where the last inequality uses $\lambda = 33$.

For the other direction, similar to Eq. (20), we know that

$$\mathrm{Reg}_m(\pi) - \mathrm{Reg}(\pi)$$
$$= (R_m(\pi_{f_m}) - R_m(\pi)) - (R(\pi_{f^\star}) - R(\pi))$$
$$\leq (R(\pi_{f_m}) - R(\pi)) - (R(\pi_{f_m}) - R(\pi))$$
$$\leq |R_m(\pi_{f_m}) - R(\pi_{f_m})| + |R_m(\pi) - R(\pi)|$$
$$\leq 2(K+1)^2\sqrt{V(q_{m-1},\pi_{f_m})\mathbf{Err}_{\log}(2^{m-2},1/T^2,\mathcal{F})}$$
$$\quad + 2(K+1)^2\sqrt{V(q_{m-1},\pi)\mathbf{Err}_{\log}(2^{m-2},1/T^2,\mathcal{F})}$$
$$\leq \frac{(K+1)^4V(q_{m-1},\pi_{f_m})}{8\gamma_m} + \frac{(K+1)^4V(q_{m-1},\pi)}{8\gamma_m} + 16\gamma_m\mathbf{Err}_{\log}(2^{m-2},1/T^2,\mathcal{F})$$
$$\text{(AM-GM inequality)}$$
$$\overset{(i)}{=} \frac{(K+1)^4V(q_{m-1},\pi_{f_m})}{8\gamma_m} + \frac{(K+1)^4V(q_{m-1},\pi)}{8\gamma_m} + \frac{16N(K+1)^4}{\gamma_m}, \tag{23}$$

where $(i)$ is again because $\gamma_m = \sqrt{\frac{N(K+1)^4}{\mathbf{Err}_{\log}(2^{m-2},1/T^2,\mathcal{F})}}$. Applying Eq. (21) to the first term in Eq. (23), we know that

$$\frac{(K+1)^4V(q_{m-1},\pi_{f_m})}{8\gamma_m}$$
$$\leq \frac{N(K+1)^4}{8\gamma_m} + \frac{\gamma_{m-1}\mathrm{Reg}_{m-1}(\pi_{f_m})}{8\gamma_m}$$
$$\leq \frac{N(K+1)^4}{8\gamma_m} + \frac{\gamma_{m-1}\left(2\mathrm{Reg}(\pi_{f_m}) + \frac{\lambda N(K+1)^4}{\gamma_{m-1}}\right)}{8\gamma_m}$$
$$\overset{(i)}{\leq} \frac{\lambda+1}{8\gamma_m} \cdot N(K+1)^4 + \frac{1}{4}\left(2\mathrm{Reg}_m(\pi_{f_m}) + \frac{\lambda N(K+1)^4}{\gamma_m}\right)$$
$$\overset{(ii)}{=} \frac{1+3\lambda}{8\gamma_m}N(K+1)^4,$$

where $(i)$ is because $\gamma_{m-1} \leq \gamma_m$ and Eq. (22), and $(ii)$ is due to $\mathrm{Reg}_m(\pi_{f_m}) = 0$. Plugging the above back to Eq. (23), we obtain that

$$\mathrm{Reg}_m(\pi) \leq \mathrm{Reg}(\pi) + \frac{2+4\lambda}{8\gamma_m}N(K+1)^4 + \frac{1}{4}\mathrm{Reg}(\pi) + \frac{16N(K+1)^4}{\gamma_m}$$

$$\leq 2\mathrm{Reg}(\pi) + \frac{\lambda N(K+1)^4}{\gamma_m}, \qquad \text{(since } \lambda = 33)$$

which finishes the proof. □ Now we are ready to prove Theorem 3.7.

**Algorithm 2** Contextual MNL Algorithms via an Online Regression Oracle

---

Input: an online regression oracle $\mathsf{Alg}_{\mathrm{on}}$ satisfying Assumption 3.
**for** $t = 1, 2, \ldots, T$ **do**

    Obtain value predictor $f_t$ from oracle $\mathsf{Alg}_{\mathrm{on}}$.
    Receive context $x_t \in \mathcal{X}$ and reward vector $r_t \in [0, 1]^N$.
    Calculate $q_t \in \Delta(\mathcal{S})$ based on $f(x_t)$ and $r_t$, via either Eq. (9) or Eq. (10).
    Sample $S_t \sim q_t$ and receive purchase decision $i_t \in S_t \cup \{0\}$ drawn according Eq. (1).
    Feed the tuple $(x_t, S_t, i_t)$ to the oracle $\mathsf{Alg}_{\mathrm{on}}$.

---

**Theorem 3.7** *Under Assumption 1 and Assumption 2, Algorithm 1 with $q_m$ defined in Eq. (5) and the optimal choice of $\gamma_m$ ensures* $\mathbf{Reg}_{\mathsf{MNL}} = \mathcal{O}\left( \sum_{m=1}^{\lceil \log_2 T \rceil} 2^m K^2 \sqrt{N \mathbf{Err}_{\log}(2^{m-1}, 1/T^2, \mathcal{F})} \right)$.

**Proof** Choose $\gamma_m = \max\left\{ 1, \sqrt{\frac{N(K+1)^4}{\mathbf{Err}_{\log}(2^{m-2}, 1/T^2, \mathcal{F})}} \right\}$ for all $m \geq 2$. Consider the regret within epoch $m \geq 2$. We first show that $\sum_{\pi \in \Psi} Q_m(\pi) \mathrm{Reg}_m(\pi) \leq \frac{N(K+1)^4}{\gamma_m}$. Concretely, according to Lemma 3.6 and Lemma B.5, we know that

$$\sum_{\pi \in \Psi} Q_m(\pi) \mathrm{Reg}_m(\pi)$$

$$= \mathbb{E}_{(x,r) \sim \mathcal{D}} \left[ \sum_{S \in \mathcal{S}} q_m(S|x, r) \left( \max_{S^\star \in \mathcal{S}} R(S^\star, f_m(x), r) - R(S, f_m(x), r) \right) \right] \leq \frac{N(K+1)^4}{\gamma_m}. \quad (24)$$

Now consider the regret within epoch $m$. Since Event 1 holds with probability at least $1 - \frac{1}{T}$, we know that

$$\mathbb{E}\left[ \sum_{t=\tau_m+1}^{\tau_{m+1}} \left( \max_{S \in \mathcal{S}} R(S, x_t, f^\star(x_t)) - R(S_t, x_t, f^\star(x_t)) \right) \right]$$

$$= (\tau_{m+1} - \tau_m) \mathbb{E}\left[ \sum_{\pi \in \Psi} Q_m(\pi) \mathrm{Reg}(\pi) \right]$$

$$\leq \frac{\tau_{m+1} - \tau_m}{T} + (\tau_{m+1} - \tau_m) \mathbb{E}\left[ \sum_{\pi \in \Psi} Q_m(\pi) \mathrm{Reg}(\pi) \;\middle|\; \text{Event 1 holds} \right]$$

$$\text{(since Event 1 holds with probability at least } 1 - \tfrac{1}{T})$$

$$\overset{(i)}{\leq} \frac{\tau_{m+1} - \tau_m}{T} + (\tau_{m+1} - \tau_m) \mathbb{E}\left[ \sum_{\pi \in \Psi} Q_m(\pi) \left( 2\mathrm{Reg}_m(\pi) + \frac{33N(K+1)^4}{\gamma_m} \right) \;\middle|\; \text{Event 1 holds} \right]$$

$$\leq \frac{\tau_{m+1} - \tau_m}{T} + (\tau_{m+1} - \tau_m) \cdot \frac{35N(K+1)^4}{\gamma_m} \qquad \text{(using Eq. (24))}$$

$$= \mathcal{O}\left( \frac{\tau_{m+1} - \tau_m}{T} + 2^{m-1} K^2 \sqrt{N \mathbf{Err}_{\log}(2^{m-2}, 1/T^2, \mathcal{F})} \right),$$

where $(i)$ uses Lemma B.8. Taking summation over $m = 2, 3, \ldots, \lceil \log_2 T + 1 \rceil$, we conclude that

$$\mathbf{Reg}_{\mathsf{MNL}} = \mathcal{O}\left( \sum_{m=1}^{\lceil \log_2 T \rceil} 2^m K^2 \sqrt{N \mathbf{Err}_{\log}(2^{m-1}, 1/T^2, \mathcal{F})} \right).$$

$\square$

# C  Omitted Details in Section 4.1

In this section, we show omitted details in Section 4.1.

### C.1 Online Regression Oracle

We first show that there exists efficient online regression oracle for the finite class and the linear class.

**Lemma C.1** *For the finite class and the linear class discussed in Lemma 3.1, the following concrete oracles satisfy Assumption 3:*

- *(Finite class) Hedge [16] with* $\mathbf{Reg}_{\log}(T, \mathcal{F}) = \mathcal{O}(\sqrt{T \log |\mathcal{F}|} \log \frac{K}{\beta})$;

- *(Linear class) Online Gradient Descent [32] with* $\mathbf{Reg}_{\log}(T, \mathcal{F}) = \mathcal{O}(B\sqrt{T})$.

**Proof** We first consider the finite function class. Since for any $S \in \mathcal{S}$, $i \in S \cup \{0\}$, and $x \in \mathcal{X}$, we have $f_i(x) \geq \beta$, we know that $\ell_{\log}(\mu(S, f(x)), i) \leq \log \frac{K+1}{\beta}$. Therefore, Hedge [16] guarantees that $\mathbf{Reg}_{\log}(T, \mathcal{F}) = \mathcal{O}\left(\log \frac{K}{\beta} \sqrt{T \log |\mathcal{F}|}\right)$.

For the linear class, we first prove that given $S \in \mathcal{S}$, $i \in S \cup \{0\}$ and $x \in \mathbb{R}^{d \times N}$, for any $f_\theta \in \mathcal{F}$, $\ell_{\log}(\mu(S, f_\theta(x)), i)$ is convex in $\theta$. Specifically, for $u \in \mathbb{R}^d$, $h(u) = \log(\sum_{i=1}^d e^{u_i})$ is convex in $u$ since for any $\alpha \in \mathbb{R}^d$,

$$
\alpha^\top \nabla_u^2 h(u) \alpha = \alpha^\top \left( \frac{1}{\mathbf{1}^\top u} \mathrm{diag}(u) - \frac{1}{(\mathbf{1}^\top u)^2} u u^\top \right) \alpha
$$
$$
= \frac{(\sum_{k=1}^d u_k \alpha_k^2)(\sum_{k=1}^d u_k) - (\sum_{k=1}^d u_k \alpha_k)^2}{(\mathbf{1}^\top u)^2} \geq 0,
$$

where the last inequality is due to Cauchy-Schwarz inequality. Define $x_0 = \mathbf{0} \in \mathbb{R}^d$ to be the $d$-dimensional all-zero vector. Then, we know that $\ell_{\log}(\mu(S, f_\theta(x), i)) = \log\left(e^{\theta^\top x_0} + \sum_{j \in S} e^{\theta^\top x_j - B}\right) - (\theta^\top x_i - B) \cdot \mathbb{1}\{i \neq 0\}$ is convex in $\theta$. Moreover, direct calculation shows that

$$
\|\nabla_\theta \ell_{\log}(\mu(S, f_\theta(x)), i)\|_2 = \left\| \frac{\sum_{j \in S} e^{\theta^\top x_j - B} \cdot x_j}{1 + \sum_{j \in S} e^{\theta^\top x_j - B}} - x_i \cdot \mathbb{1}\{i \neq 0\} \right\|_2 \leq 2.
$$

Therefore, Online Gradient Descent [32] guarantees that $\mathbf{Reg}_{\log}(T, \mathcal{F}) = \mathcal{O}(B\sqrt{T})$, since $\|\theta\|_2 \leq B$. $\qquad \square$

For completeness, we restate and prove Lemma 4.2, which is extended from the analysis in [14, 15].

**Lemma C.2** *Under Assumption 1 and Assumption 3, Algorithm 2 (with any $q_t$) ensures*

$$
\mathbf{Reg}_{\mathrm{MNL}} \leq \mathbb{E}\left[ \sum_{t=1}^T \mathsf{dec}_\gamma(q_t; f_t(x_t), r_t) \right] + 2\gamma \mathbf{Reg}_{\log}(T, \mathcal{F})
$$

*for any $\gamma > 0$, where $\mathsf{dec}_\gamma(q; v, r)$ is the* Decision-Estimation Coefficient (DEC) *defined as*

$$
\max_{v^\star \in [0,1]^N} \max_{S^\star \in \mathcal{S}} \left\{ R(S^\star, v^\star, r) - \mathbb{E}_{S \sim q}\left[R(S, v^\star, r)\right] - \gamma \mathbb{E}_{S \sim q}\left[ \|\mu(S, v) - \mu(S, v^\star)\|_2^2 \right] \right\}. \quad (8)
$$

**Proof** Following the regret decomposition in [14, 15], we decompose $\mathbf{Reg}_{\mathrm{MNL}}$ as follows:

$$
\mathbf{Reg}_{\mathrm{MNL}}
$$
$$
= \mathbb{E}\left[ \sum_{t=1}^T \max_{S^\star \in \mathcal{S}} R(S, f^\star(x_t), r_t) - \sum_{t=1}^T q_t(S) R(S, f^\star(x_t), r_t) \right]
$$
$$
= \mathbb{E}\left[ \sum_{t=1}^T \max_{S^\star \in \mathcal{S}} R(S^\star, f^\star(x_t), r_t) - \sum_{t=1}^T q_t(S) R(S, f^\star(x_t), r_t) \right.
$$
$$
\left. - \gamma \sum_{S \in \mathcal{S}} q_t(S) \|\mu(S, f_t(x_t)) - \mu(S, f^\star(x_t))\|_2^2 \right]
$$

$$+ \gamma \mathbb{E}\left[\sum_{S \in \mathcal{S}} q_t(S)\|\mu(S, f_t(x_t)) - \mu(S, f^\star(x_t))\|_2^2\right]$$

$$\leq \mathbb{E}\left[\sum_{t=1}^{T} \max_{S^\star \in \mathcal{S}, v^\star \in [0,1]^N} \left\{ R(S^\star, v^\star, r_t) - \sum_{t=1}^{T} q_t(S)R(S, v^\star, r_t) - \right.\right.$$

$$\left.\left. \gamma \sum_{S \in \mathcal{S}} q_t(S)\|\mu(S, f_t(x_t)) - \mu(S, v^\star)\|_2^2 \right\}\right]$$

$$+ \gamma \cdot \mathbb{E}\left[\sum_{t=1}^{T} \|\mu(S_t, f_t(x_t)) - \mu(S_t, f^\star(x_t))\|_2^2\right]$$

$$= \mathbb{E}\left[\sum_{t=1}^{T} \mathsf{dec}_\gamma(q_t; f_t(x_t), r_t)\right] + \gamma \cdot \mathbb{E}\left[\sum_{t=1}^{T} \|\mu(S_t, f_t(x_t)) - \mu(S_t, f^\star(x_t))\|_2^2\right], \quad (25)$$

where the last equality is by the definition of $\mathsf{dec}_\gamma(q_t; f_t(x_t), r_t)$. According to Lemma 3.3, we know that

$$\mathbb{E}\left[\sum_{t=1}^{T} \|\mu(S_t, f_t(x_t)) - \mu(S_t, f^\star(x_t))\|_2^2\right]$$

$$\leq 2\mathbb{E}\left[\sum_{t=1}^{T} \ell_{\log}(\mu(S_t, f_t(x_t)), i_t) - \sum_{t=1}^{T} \ell_{\log}(\mu(S_t, f^\star(x_t)), i_t)\right] \leq 2\mathbf{Reg}_{\log}(T, \mathcal{F}). \quad (26)$$

Combining Eq. (25) and Eq. (26) finishes the proof. $\qquad\square$

## C.2  Proof of Theorem 4.3

Next, we prove Theorem 4.3, which shows that similar to the stochastic environment, a simple but efficient $\varepsilon$-greedy strategy achieves $\mathcal{O}\left(T^{2/3}(NK\mathbf{Reg}_{\log}(T, \mathcal{F}))^{1/3}\right)$ expected regret.

**Theorem 4.3** *The strategy defined in Eq. (9) guarantees $\mathsf{dec}_\gamma(q_t; f_t(x_t), r_t) = \mathcal{O}(\frac{NK}{\gamma\varepsilon} + \varepsilon)$. Consequently, under Assumption 1 and Assumption 3, Algorithm 2 with $q_t$ calculated via Eq. (9) and the optimal choice of $\varepsilon$ and $\gamma$ ensures $\mathbf{Reg}_{\mathsf{MNL}} = \mathcal{O}\left((NK\mathbf{Reg}_{\log}(T, \mathcal{F}))^{\frac{1}{3}} T^{\frac{2}{3}}\right)$.*

**Proof** We first prove that $q_t$ defined in Eq. (9) guarantees $\mathsf{dec}_\gamma(q_t; f_t(x_t), r_t) \leq \mathcal{O}\left(\frac{NK}{\gamma\varepsilon} + \varepsilon\right)$. Specifically, for any $S^\star \in \mathcal{S}$ and $v^\star \in [0, 1]^N$, we know that

$$R(S^\star, v^\star, r_t) - \sum_{S \in \mathcal{S}} q_t(S)R(S, v^\star, r_t) - \gamma \sum_{S \in \mathcal{S}} q_t(S)\|\mu(S, f_t(x_t)) - \mu(S, f^\star(x_t))\|_2^2$$

$$\overset{(i)}{\leq} \sum_{i \in S^\star} |v_i^\star - f_{t,i}(x_t)| + \sum_{S \in \mathcal{S}} q_t(S) \sum_{i \in S} |\mu_i(S, v_i^\star) - \mu_i(S, f_t(x_t))|$$

$$+ R(S^\star, f_t(x_t), r_t) - \sum_{S \in \mathcal{S}} q_t(S)R(S, f_t(x_t), r_t) - \gamma \sum_{S \in \mathcal{S}} q_t(S)\|\mu(S, f_t(x_t)) - \mu(S, v^\star)\|_2^2$$

$$\overset{(ii)}{\leq} \sum_{i \in S^\star} |v_i^\star - f_{t,i}(x_t)| + \frac{2K}{\gamma}$$

$$+ R(S^\star, f_t(x_t), r_t) - \sum_{S \in \mathcal{S}} q_t(S)R(S, f_t(x_t), r_t) - \frac{\gamma}{2} \sum_{S \in \mathcal{S}} q_t(S)\|\mu(S, f_t(x_t)) - \mu(S, v^\star)\|_2^2$$

$$\overset{(iii)}{\leq} \sum_{i \in S^\star} |v_i^\star - f_{t,i}(x_t)| + \frac{2K}{\gamma} + \varepsilon + R(S^\star, f_t(x_t), r_t) - \max_{S \in \mathcal{S}} R(S, f_t(x_t), r_t)$$

$$- \frac{\gamma\varepsilon}{2N} \sum_{i=1}^{N} \|\mu(\{i\}, f_t(x_t)) - \mu(\{i\}, v^\star)\|_2^2$$

$$\overset{(iv)}{\leq} \sum_{i \in S^\star} |v_i^\star - f_{t,i}(x_t)| + \frac{2K}{\gamma} + \varepsilon + R(S^\star, f_t(x_t), r_t) - \max_{S \in \mathcal{S}} R(S, f_t(x_t), r_t)$$

$$- \frac{\gamma \varepsilon}{64N} \sum_{i=1}^{N} (f_{t,i}(x_t) - v_i^\star)^2$$

$$\overset{(v)}{\leq} \frac{16NK}{\gamma \varepsilon} + \frac{2K}{\gamma} + \varepsilon$$

$$\leq \mathcal{O}\left(\frac{NK}{\gamma \varepsilon} + \varepsilon\right),$$

where $(i)$ uses Lemma B.1, $(ii)$ is due to AM-GM inequality and $|S| \leq K$, $(iii)$ is according to the construction of $q_t$ and $R(S, v, r) \in [0, 1]$, $(iv)$ uses Lemma B.3 with $d = 1$, and $(v)$ is uses AM-GM inequality and the fact that $|S^\star| \leq K$. Taking maximum over all $S^\star \in \mathcal{S}$ and $v^\star \in [0, 1]^N$ proves that $\mathsf{dec}_\gamma(q_t; f_t(x_t), r_t) \leq \mathcal{O}\left(\frac{NK}{\gamma \varepsilon} + \varepsilon\right)$.

Combining the above result with Lemma 4.2, we know that

$$\mathbf{Reg}_{\mathsf{MNL}} = \mathcal{O}\left(\frac{NKT}{\gamma \varepsilon} + \varepsilon T + \gamma \mathbf{Reg}_{\log}(T, \mathcal{F})\right).$$

Picking $\gamma$ and $\varepsilon$ optimally finishes the proof. $\qquad \square$

## C.3  Proof of Theorem 4.5 and Theorem 4.6

In this section, we restate and prove Theorem 4.5, which proves that $q_t$ calculated via Eq. (10) guarantees that $\mathsf{dec}_\gamma(q_t; f_t(x_t), r_t) \leq \mathcal{O}\left(\frac{NK^4}{\gamma}\right)$.

**Theorem 4.5** *The following distribution satisfies* $\mathsf{dec}_\gamma(q_t, f_t(x_t), r_t) \leq \mathcal{O}\left(\frac{NK^4}{\gamma}\right)$:

$$q_t = \operatorname*{argmax}_{q \in \Delta(\mathcal{S})} \mathbb{E}_{S \sim q}\left[R(S, f_t(x_t), r_t)\right] - \frac{(K+1)^4}{\gamma} \sum_{i=1}^{N} \log \frac{1}{w_i(q)}. \tag{10}$$

**Proof**  Since the construction of $q_t$ is the same as Eq. (5) with $f_m$ replaced by $f_t$ and $\gamma_m$ replaced by $\gamma$, according to Lemma 3.6, we know that $q_t$ satisfies that

$$\max_{S^\star \in \mathcal{S}} R(S^\star, f_t(x_t), r_t) - \sum_{S \in \mathcal{S}} q_t(S) \cdot R(S, f_t(x_t), r_t) \leq \frac{N(K+1)^4}{\gamma}, \tag{27}$$

$$\forall S \in \mathcal{S}, \quad \sum_{i \in S} \frac{1}{w_i(q)} \leq N + \frac{\gamma}{(K+1)^4}\left(\max_{S^\star \in \mathcal{S}} R(S^\star, f_t(x_t), r_t) - R(S, f_t(x_t), r_t)\right). \tag{28}$$

Using Eq. (27) and Eq. (28), we know that for any $S^\star \in \mathcal{S}$ and $v^\star \in [0, 1]^N$,

$$R(S^\star, v^\star, r_t) - \sum_{S \in \mathcal{S}} q_t(S) R(S, v^\star, r_t) - \gamma \sum_{S \in \mathcal{S}} q_t(S) \|\mu(S, f_t(x_t)) - \mu(S, f^\star(x_t))\|_2^2$$

$$\leq \sum_{i \in S^\star} |v_i^\star - f_{t,i}(x_t)| + \sum_{S \in \mathcal{S}} q_t(S) \sum_{i \in S} |v_i^\star - f_{t,i}(x_t)| \qquad \text{(according to Lemma B.1)}$$

$$+ R(S^\star, f_t(x_t), r_t) - \sum_{S \in \mathcal{S}} q_t(S) R(S, f_t(x_t), r_t) - \gamma \sum_{S \in \mathcal{S}} q_t(S) \|\mu(S, f_t(x_t)) - \mu(S, v^\star)\|_2^2$$

$$\leq \sum_{i \in S^\star} |v_i^\star - f_{t,i}(x_t)| + \sum_{i=1}^{N} w_i(q_t) \cdot |v_i^\star - f_{t,i}(x_t)| \qquad \text{(by definition of } w_i(q))$$

$$+ R(S^\star, f_t(x_t), r_t) - \sum_{S \in \mathcal{S}} q_t(S) R(S, f_t(x_t), r_t) - \gamma \sum_{S \in \mathcal{S}} q_t(S) \|\mu(S, f_t(x_t)) - \mu(S, v^\star)\|_2^2.$$

$$\leq \sum_{i\in S^\star} |v_i^\star - f_{t,i}(x_t)| + \sum_{i=1}^N w_i(q_t)\cdot|v_i^\star - f_{t,i}(x_t)|$$

$$+ R(S^\star, f_t(x_t), r_t) - \sum_{S\in\mathcal{S}} q_t(S)R(S, f_t(x_t), r_t) - \frac{\gamma}{2(K+1)^4}\sum_{S\in\mathcal{S}}q_t(S)\sum_{i\in S}(v_i^\star - f_{t,i}(x_t))^2$$
$$\text{(according to Lemma 3.3)}$$

$$= \sum_{i\in S^\star} |v_i^\star - f_{t,i}(x_t)| + \sum_{i=1}^N w_i(q_t)\cdot|v_i^\star - f_{t,i}(x_t)|$$

$$+ R(S^\star, f_t(x_t), r_t) - \sum_{S\in\mathcal{S}} q_t(S)R(S, f_t(x_t), r_t) - \frac{\gamma}{2(K+1)^4}\sum_{i=1}^N w_i(q_t)(v_i^\star - f_{t,i}(x_t))^2$$

$$\leq \frac{N(K+1)^4}{\gamma} + \sum_{i\in S^\star}\frac{(K+1)^4}{\gamma w_i(q_t)} + R(S^\star, f_t(x_t), r_t) - \sum_{S\in\mathcal{S}} q_t(S)R(S, f_t(x_t), r_t)$$
$$\text{(AM-GM inequality)}$$

$$= \frac{N(K+1)^4}{\gamma} + \sum_{i\in S^\star}\frac{(K+1)^4}{\gamma w_i(q_t)} - \left(\max_{S_0\in\mathcal{S}} R(S_0, f_t(x_t), r_t) - R(S^\star, f_t(x_t), r_t)\right)$$

$$+ \max_{S_0\in\mathcal{S}} R(S_0, f_t(x_t), r_t) - \sum_{S\in\mathcal{S}} q_t(S)R(S, f_t(x_t), r_t)$$

$$\leq \frac{N(K+1)^4}{\gamma} + \frac{(K+1)^4}{\gamma}\left(N + \frac{\gamma}{(K+1)^4}\left(\max_{S_0\in\mathcal{S}} R(S_0, f_t(x_t), r_t) - R(S^\star, f_t(x_t), r_t)\right)\right)$$

$$- \left(\max_{S_0\in\mathcal{S}} R(S_0, f_t(x_t), r_t) - R(S^\star, f_t(x_t), r_t)\right) + \frac{N(K+1)^4}{\gamma}$$
$$\text{(according to Eq. (27) and Eq. (28))}$$

$$= \frac{3N(K+1)^4}{\gamma}.$$

Taking maximum over all $S^\star \in \mathcal{S}$ and $v^\star \in [0,1]^N$ finishes the proof. $\qquad\square$

Combining Lemma 4.2 and Theorem 4.5, we are able to prove Theorem 4.6.

**Theorem 4.6** *Under Assumption 1 and Assumption 3, Algorithm 2 with $q_t$ calculated via Eq. (10) and the optimal choice of $\gamma$ ensures* $\mathbf{Reg}_{\mathsf{MNL}} = \mathcal{O}\left(K^2\sqrt{NT\mathbf{Reg}_{\log}(T,\mathcal{F})}\right)$.

**Proof** Combining Lemma 4.2 and Theorem 4.5, we know that Algorithm 2 with $q_t$ calculated via Eq. (10) satisfies that $\mathbf{Reg}_{\mathsf{MNL}} = \mathcal{O}\left(\frac{NK^4}{\gamma} + \gamma\mathbf{Reg}_{\log}(T,\mathcal{F})\right)$. Picking $\gamma = K^2\sqrt{\frac{NT}{\mathbf{Reg}_{\log}(T,\mathcal{F})}}$ finishes the proof. $\qquad\square$

# D Regression Oracle for More Function Classes

In this section, we provide examples on regression oracles for a broader Lipschitz function class satisfying Assumption 2 and Assumption 3.

**Lemma D.1** *Suppose that $\mathcal{F}$ is a 1-Lipschitz function class defined as $\mathcal{F} = \{f_{\theta,i}(x)\in[\beta,1] \mid \theta\in[0,1]^d\}$ where $\beta > 0$ and $\|f_{\theta_1,i} - f_{\theta_2,i}\|_\infty \leq \|\theta_1 - \theta_2\|_\infty$ for all $\theta_1,\theta_2\in[0,1]^d$ and $i\in[N]$. Then, ERM strategy $\widehat{f}_D = \arg\min_{f\in\mathcal{F}}\sum_{(x,S,i)\in D}\ell_{\log}(\mu(S,f(x)),i)$ satisfies Assumption 2 with $\mathbf{Err}_{\log}(n,\delta,\mathcal{F}) = \mathcal{O}\left(\frac{d\log\frac{K}{\beta}\log\frac{n}{\beta}\log\frac{1}{\delta}}{n}\right)$. Moreover, there exists an algorithm satisfying Assumption 3 with $\mathbf{Reg}_{\log}(T,\mathcal{F}) = \mathcal{O}\left(\sqrt{dT\log(T/\beta)}\log(K/\beta)\right)$.*

**Proof** We first consider the ERM strategy. For notational convenience, let $Z \triangleq (x,S,i)$ and with an abuse of notation, we denote $\ell_{\log}(\mu(S,f(x)),i)$ by $\ell_{\log}^f(Z)$. According to Theorem 7.7 in [26],

we know that for any $\mathcal{F} = \{f : \mathcal{X} \mapsto [\beta, 1]^N\}$ such that the $\varepsilon$-covering number of $\{\ell_{\log}^f : f \in \mathcal{F}\}$ is $\mathcal{N}(\varepsilon)$, ERM predictor $\widehat{f}_D$ guarantees that with probability $1 - \delta$:

$$\mathbb{E}_{Z \sim \mathcal{D}}\left[\ell_{\log}^{\widehat{f}_D}(Z)\right] \leq \mathbb{E}_{Z \sim \mathcal{D}}\left[\ell_{\log}^{f^\star}(Z)\right] + \mathcal{O}\left(\frac{\log \frac{K}{\beta} \log(\mathcal{N}(\frac{1}{n^2})) \log \frac{1}{\delta}}{n}\right). \tag{29}$$

Now we show that for the 1-Lipschitz function class, $\mathcal{N}(\varepsilon) \leq \left(1 + \frac{2}{\beta\varepsilon}\right)^d$. Specifically, define the $\frac{\beta\varepsilon}{2}$-grid of $[0,1]^d$ as $\mathcal{C}(\varepsilon) = \{\theta \in [0,1]^d : \theta_i \in \{0, \frac{\beta\varepsilon}{2}, \beta\varepsilon, \ldots, 1\}, i \in [N]\}$. For any $\theta_1 \in [0,1]^d$, let $\theta_2 = \mathrm{argmin}_{\theta \in \mathcal{C}(\varepsilon)} \|\theta - \theta_1\|_\infty$. By definition, we know that $\|\theta_1 - \theta_2\|_\infty \leq \frac{\beta\varepsilon}{2}$. Given any $Z = (x, S, i)$,

$$\left|\ell_{\log}^{f_{\theta_1}}(Z) - \ell_{\log}^{f_{\theta_2}}(Z)\right|$$

$$= |\ell_{\log}(\mu(S, f_{\theta_1}(x)), i) - \ell_{\log}(\mu(S, f_{\theta_2}(x)), i)|$$

$$\leq \left|\log \frac{1 + \sum_{j \in S} f_{\theta_1, j}(x)}{1 + \sum_{j \in S} f_{\theta_2, j}(x)}\right| + \left|\log \frac{f_{\theta_1, i}(x)}{f_{\theta_2, i}(x)} \cdot \mathbb{1}\{i \neq 0\}\right|$$

$$= \log \frac{1 + \max\{\sum_{j \in S} f_{\theta_1, j}(x), \sum_{j \in S} f_{\theta_2, j}(x)\}}{1 + \min\{\sum_{j \in S} f_{\theta_1, j}(x), \sum_{j \in S} f_{\theta_2, j}(x)\}} + \log \frac{\max\{f_{\theta_1, i}(x), f_{\theta_2, i}(x)\}}{\min\{f_{\theta_1, i}(x), f_{\theta_2, i}(x)\}} \cdot \mathbb{1}\{i \neq 0\}$$

$$= \log\left(1 + \frac{\left|\sum_{j \in S} f_{\theta_1, j}(x) - \sum_{j \in S} f_{\theta_2, j}(x)\right|}{1 + \min\{\sum_{j \in S} f_{\theta_1, j}(x), \sum_{j \in S} f_{\theta_2, j}(x)\}}\right)$$

$$\quad + \log\left(1 + \frac{|f_{\theta_1, i}(x) - f_{\theta_2, i}(x)|}{\min\{f_{\theta_1, i}(x), f_{\theta_2, i}(x)\}}\right) \cdot \mathbb{1}\{i \neq 0\}$$

$$\leq \sum_{j \in S} \frac{|f_{\theta_1, j}(x) - f_{\theta_2, j}(x)|}{1 + \min\{\sum_{j \in S} f_{\theta_1, j}(x), \sum_{j \in S} f_{\theta_2, j}(x)\}} + \mathbb{1}\{i \neq 0\} \frac{|f_{\theta_1, i}(x) - f_{\theta_2, i}(x)|}{\min\{f_{\theta_2, i}(x), f_{\theta_1, i}(x)\}}$$

$$\leq \frac{1}{2} \frac{|S|\beta\varepsilon}{1 + \beta|S|} + \frac{\beta\varepsilon}{2\beta}$$

$$\leq \varepsilon, \tag{30}$$

where the second inequality is by $\log(1 + x) \leq x$ for $x \geq 0$ and the triangular inequality, and the third inequality is due to the Lipschitz property and the lower bound $\beta$ on all values. Therefore, we know that $\mathcal{N}(\varepsilon) \leq |\mathcal{C}(\varepsilon)| \leq \left(1 + \frac{2}{\beta\varepsilon}\right)^d$. Plugging in this to Eq. (29) proves the claim.

Next, we consider the adversarial environment. Consider applying Hedge [16] on the discretized set $\mathcal{C}(1/T)$. Since $\ell_{\log}(\mu(S, f(x)), i) \leq \log \frac{K+1}{\beta}$ for all context $x$, $S \in \mathcal{S}$, and $i \in S \cup \{0\}$, Hedge guarantees that

$$\mathbb{E}\left[\sum_{t=1}^T \ell_{\log}(\mu(S_t, f_{\theta_t}(x_t)), i_t) - \sum_{t=1}^T \ell_{\log}(\mu(S_t, f_{\widehat{\theta}}(x_t)), i_t)\right]$$

$$\leq \mathcal{O}\left(\log \frac{K}{\beta} \sqrt{T \log |\mathcal{C}(1/T)|}\right)$$

$$= \mathcal{O}\left(\log \frac{K}{\beta} \sqrt{dT \log(T/\beta)}\right),$$

for all $\widehat{\theta} \in \mathcal{C}(1/T)$. Picking $\widehat{\theta} = \mathrm{argmin}_{\theta \in \mathcal{C}(1/T)} \|\theta - \theta^\star\|_\infty$ where $f^\star \triangleq f_{\theta^\star}$ and applying Eq. (30) to $\widehat{\theta}$ and $\theta^\star$ finishes the proof. $\qquad\square$

# E    Omitted Details in Section 4.2

In this section, we show omitted details in Section 4.2. We start by describing the algorithm: it maintains a distribution $p_t$ over the value function class $\mathcal{F}$, and at each round $t$, it samples $f_t$ from

---

**Algorithm 3** Feel-Good Thompson Sampling for Contextual MNL bandits

---

Input: a learning rate $\eta > 0$.
Initialize $p_1 \in \Delta(\mathcal{F})$ to be the uniform distribution over $\mathcal{F}$.
**for** $t = 1, 2, \ldots, T$ **do**

    Sample a value function $f_t$ from $p_t$.
    Receive context $x_t$ and reward vector $r_t \in [0, 1]^N$.
    Select $S_t = \text{argmax}_{S \in \mathcal{S}} R(S, f_t(x), r_t)$ and receive feedback $i_t \in S_t \cup \{0\}$.
    Define the loss estimator $\widehat{\ell}_{t,f}$ for each $f \in \mathcal{F}$ as

$$\widehat{\ell}_{t,f} = \frac{1}{16\eta} \sum_{i \in S_t} (\mu_i(S_t, f(x_t)) - \mathbb{1}[i = i_t])^2 - \max_{S \in \mathcal{S}} R(S, f(x_t), r_t). \tag{31}$$

    Update $p_{t+1,f} \propto p_{t,f} \cdot \exp(-\eta \widehat{\ell}_{t,f})$.

---

$p_t$ and selects the subset $S_t$ that maximizes the expected reward with respect to the value function $f_t$ and the reward vector $r_t$. After receiving the purchase decision $i_t$, the algorithm constructs a loss estimator $\widehat{\ell}_{t,f}$ for each $f \in \mathcal{F}$ as defined in Eq. (31), and updates the distribution $p_t$ using a standard multiplicative update with learning rate $\eta$. See Algorithm 3.

The idea of the loss estimator Eq. (31) is as follows. The first term measures how accurate $f$ is via the squared distance between the multinomial distribution induced by $f$ and the true outcome. The second term, which is the highest expected reward one could get if the value function was $f$, is subtracted from the first term to serve as a form of optimism (the "feel-good" part), encouraging exploration for those $f$'s that promise a high reward.

We extend the analysis of Zhang [30] and combine it with our technical lemmas (such as Lemma 3.3 and Lemma B.1) to prove the following regret guarantee, where the term $Z_T$ should be interpreted as a certain complexity measure for the class $\mathcal{F}$.

**Theorem E.1** *Under Assumption 1, Algorithm 3 with learning rate $\eta \leq 1$ ensures* $\mathbf{Reg}_{\text{MNL}} \leq 32\eta N(K+1)^4 T + 4\eta T + \frac{Z_T}{\eta}$, *where* $Z_T = -\mathbb{E}[\log \mathbb{E}_{f \sim p_1}[\exp(-\eta \sum_{t=1}^{T} (\widehat{\ell}_{t,f} - \widehat{\ell}_{t,f^\star}))]]$.

**Proof** First, we decompose the regret as follows:

$$\mathbf{Reg}_{\text{MNL}} = \mathbb{E}\left[\sum_{t=1}^{T}(\max_{S \in \mathcal{S}} R(S, f^\star(x_t), r_t) - R(S_t, f^\star(x_t), r_t))\right]$$

$$= \mathbb{E}\left[\sum_{t=1}^{T}(R(S_t, f_t(x_t), r_t) - R(S_t, f^\star(x_t), r_t))\right]$$

$$- \mathbb{E}\left[\sum_{t=1}^{T}(R(S_t, f_t(x_t), r_t) - \max_{S \in \mathcal{S}} R(S, f^\star(x_t), r_t))\right]$$

$$\overset{(i)}{=} \mathbb{E}\left[\sum_{t=1}^{T}(R(S_t, f_t(x_t), r_t) - R(S_t, f^\star(x_t), r_t))\right]$$

$$- \mathbb{E}\left[\sum_{t=1}^{T}\underbrace{(\max_{S \in \mathcal{S}} R(S, f_t(x_t), r_t) - \max_{S \in \mathcal{S}} R(S, f^\star(x_t), r_t))}_{\triangleq \text{FG}_t}\right]$$

$$\overset{(ii)}{\leq} \mathbb{E}\left[\sum_{t=1}^{T}\sum_{i \in S_t}|f_{t,i}(x_t) - f_i^\star(x_t)|\right] - \mathbb{E}\left[\sum_{t=1}^{T}\text{FG}_t\right]. \tag{32}$$

where $(i)$ is because $S_t = \text{argmax}_{S \in \mathcal{S}} R(S, f_t(x_t), r_t)$ according to Algorithm 3 and $(ii)$ is using Lemma B.1. Here, "Feel-Good" term $\text{FG}_t$ measures the difference between the expected reward of the best subset given the value predictor $f_t$ and that of the true value predictor $f^\star$.

Next, we analyze the first term $\sum_{t=1}^{T} \sum_{i \in S_t} |f_{t,i}(x_t) - f_i^\star(x_t)|$. Given any context $x \in \mathcal{X}$, reward vector $r_t \in [0,1]^N$, and a value predictor $f \in \mathcal{F}$, let $S(f(x), r) = \operatorname{argmax}_{S \in \mathcal{S}} R(S, f(x), r)$. According to Algorithm 3, we have $S_t = S(\theta_t, x_t)$. With a slight abuse of notation, for distribution $p_t$ over $\mathcal{F}$, let $w_{t,i} = \mathbb{E}_{f \sim p_t}[\mathbb{1}\{i \in S(f(x_t), r_t)\}]$ be the probability that item $i$ is included in the selected set at round $t$. Let $q_t \in \Delta(\mathcal{S})$ be the distribution over $\mathcal{S}$ induced by $p_t$, meaning that $q_t(S) = \mathbb{E}_{f \sim p_t}[\mathbb{1}\{S(f(x_t), r_t) = S\}]$. Then, for each $i \in [N]$, for any $\mu > 0$,

$$\mathbb{E}_{f \sim p_t}\left[|f_i(x_t) - f_i^\star(x_t)| \cdot \mathbb{1}\{i \in S(f(x_t), r_t)\}\right]$$

$$\leq \mathbb{E}_{f \sim p_t}\left[\frac{\mathbb{1}\{i \in S(f(x_t), r_t)\}}{4\mu w_{t,i}} + w_{t,i}(f_i(x_t) - f_i^\star(x_t))^2\right] \qquad \text{(AM-GM inequality)}$$

$$= \frac{1}{4\mu} + \mu w_{t,i} \mathbb{E}_{f \sim p_t}\left[(f_i(x_t) - f_i^\star(x_t))^2\right]. \tag{33}$$

Taking a summation over all $i \in [N]$, we know that for any $\mu > 0$,

$$\mathbb{E}\left[\sum_{i \in S_t} |f_{t,i}(x_t) - f_i^\star(x_t)|\right]$$

$$= \mathbb{E}\left[\sum_{i=1}^{N} |f_{t,i}(x_t) - f_i^\star(x_t)| \cdot \mathbb{1}\{i \in S(f_t(x_t), r_t)\}\right]$$

$$= \mathbb{E}\left[\sum_{i=1}^{N} \mathbb{E}_{f \sim p_t}\left[|f_i(x_t) - f_i^\star(x_t)| \cdot \mathbb{1}\{i \in S(f(x_t), r_t)\}\right]\right]$$

$$\overset{(i)}{\leq} \frac{N}{4\mu} + \mu \mathbb{E}\left[w_{t,i} \mathbb{E}_{f \sim p_t}\left[\sum_{i=1}^{N} (f_i(x_t) - f_i^\star(x_t))^2\right]\right]$$

$$\overset{(ii)}{=} \frac{N}{4\mu} + \mu \mathbb{E}_{S_t \sim q_t} \mathbb{E}_{f \sim p_t}\left[\sum_{i \in S_t} (f_i(x_t) - f_i^\star(x_t))^2\right], \tag{34}$$

where $(i)$ uses Eq. (33) and $(ii)$ is by definition of $w_{t,i}$ and $q_t$.

Let $\text{LS}_t = \sum_{i \in S_t} (f_i(x_t) - f_i^\star(x_t))^2$ ("Least Squares"). Combining Eq. (32) with Eq. (34), we know that

$$\mathbf{Reg}_{\text{MNL}} \leq \frac{NT}{4\mu} + \mu \mathbb{E}\left[\sum_{t=1}^{T} \mathbb{E}_{S_t \sim q_t} \mathbb{E}_{f \sim p_t}[\text{LS}_t]\right] - \mathbb{E}\left[\sum_{t=1}^{T} \text{FG}_t\right]. \tag{35}$$

To bound the last two terms in Eq. (35), using Lemma B.3 and the fact that $i_t$ is a drawn from the distribution $\mu(S_t, f^\star(x_t), r_t)$ and, we show in Lemma E.2 that

$$\frac{1}{128\eta(K+1)^4} \mathbb{E}_{f \sim p_t}[\text{LS}_t] - \mathbb{E}_{f_t \sim q_t}[\text{FG}_t] \leq -\frac{1}{\eta} \log \mathbb{E}_{i_t | x_t, S_t} \mathbb{E}_{f \sim p_t}\left[\exp(-\eta(\widehat{\ell}_{t,f} - \widehat{\ell}_{t,f^\star}))\right] + 4\eta. \tag{36}$$

Therefore, picking $\mu = \frac{1}{128\eta(K+1)^4}$ and combining Eq. (35) and Eq. (36), we know that

$$\mathbf{Reg}_{\text{MNL}}$$

$$\leq 32\eta N(K+1)^4 T + 4\eta T - \frac{1}{\eta} \mathbb{E}\left[\sum_{t=1}^{T} \log \mathbb{E}_{i_t | x_t, S_t} \mathbb{E}_{f \sim p_t}\left[\exp\left(-\eta\left(\widehat{\ell}_{t,f} - \widehat{\ell}_{t,f^\star}\right)\right)\right]\right] \tag{37}$$

To bound the last term in Eq. (37), we use the exponential weight update dynamic of $p_t$. Following a classic analysis of exponential weight update, we show in Lemma E.3 that

$$-\mathbb{E}\left[\log \mathbb{E}_{i_t | x_t, S_t} \mathbb{E}_{f \sim p_t}\left[\exp(-\eta(\widehat{\ell}_{t,f} - \widehat{\ell}_{t,f^*}))\right]\right] \leq Z_t - Z_{t-1}, \tag{38}$$

where $Z_t \triangleq -\mathbb{E}\left[\log \mathbb{E}_{f \sim p_1}\left[\exp\left(-\eta \sum_{\tau=1}^{t} \left(\widehat{\ell}_{t,f} - \widehat{\ell}_{t,f^*}\right)\right)\right]\right]$. Combining Eq. (37) and Eq. (38), we arrive at

$$\mathbf{Reg}_{\text{MNL}} \leq 32\eta N(K+1)^4 T + 4\eta T + \frac{1}{\eta} \sum_{t=1}^{T} (Z_t - Z_{t-1})$$

$$\leq 32\eta N(K+1)^4 T + 4\eta T + \frac{Z_T}{\eta},$$

where the last inequality uses the fact that $Z_0 = 0$. $\qquad\square$

**Lemma E.2** *Suppose that $\eta \leq 1$. For any distribution $p_t$ over $\mathcal{F}$, we have*

$$\frac{1}{128\eta(K+1)^4}\mathbb{E}_{f\sim p_t}[\mathrm{LS}_t] - \mathbb{E}_{f_t\sim q_t}[\mathrm{FG}_t] \leq -\frac{1}{\eta}\log\mathbb{E}_{i_t|x_t,S_t}\mathbb{E}_{f\sim p_t}\left[\exp(-\eta(\widehat{\ell}_{t,f} - \widehat{\ell}_{t,f^\star}))\right] + 4\eta,$$

*where $\mathrm{LS}_t$ and $\mathrm{FG}_t$ are defined in the proof of [Theorem E.1](#), and $\widehat{\ell}_{t,f}$ is defined in [Eq. (31)](#).*

**Proof** For notational convenience, define $c_{t,i} = \mathbb{1}\{i = i_t\}$ for all $i \in [N]$. Let $\varepsilon_{t,i} = c_{t,i} - \mu_i(S_t, f^\star(x_t))$ for all $i \in S_t$ and $\varepsilon_t \in \mathbb{R}^{N+1}$ is the corresponding vector. Consider the term $-\eta\left(\widehat{\ell}_{t,f} - \widehat{\ell}_{t,f^\star}\right)$ for an arbitrary $f \in \mathcal{F}$.

$$
\begin{aligned}
&- \eta\left(\widehat{\ell}_{t,f} - \widehat{\ell}_{t,f^\star}\right) \\
&= -\frac{1}{16}\sum_{i\in S_t}(\mu_i(S_t, f(x_t)) - c_{t,i})^2 + \frac{1}{8K}\sum_{i\in S_t}(\mu_i(S_t, f^\star(x_t)) - c_{t,i})^2 \\
&\quad + \eta\cdot\max_{S\in\mathcal{S}}R(S, f(x_t), r_t) - \eta\cdot\max_{S\in\mathcal{S}}R(S, f^\star(x_t), r_t) \\
&= -\frac{1}{16}\sum_{i\in S_t}(\mu_i(S_t, f(x_t)) - \mu_i(S_t, f^\star(x_t)))(2c_{t,i} - \mu_i(S_t, f(x_t)) - \mu_i(S_t, f^\star(x_t))) \\
&\quad + \eta\cdot\max_{S\in\mathcal{S}}R(S, f(x_t), r_t) - \eta\cdot\max_{S\in\mathcal{S}}R(S, f^\star(x_t), r_t) \\
&= \underbrace{-\frac{1}{16}\sum_{i\in S_t}(\mu_i(S_t, f(x_t)) - \mu_i(S_t, f^\star(x_t)))(\mu_i(S_t, f^\star(x_t)) - \mu_i(S_t, f(x_t)) + 2\varepsilon_{t,i})}_{\widehat{\mathrm{LS}}_t} \\
&\quad + \eta\mathrm{FG}_t(f),
\end{aligned}
$$

where we define the first term as $\widehat{\mathrm{LS}}_t$, which we will show later how this term is related to $\mathrm{LS}_t$, and the second term $\mathrm{FG}_t(f) = \max_{S\in\mathcal{S}}R(S, f(x_t), r_t) - \max_{S\in\mathcal{S}}R(S, f^\star(x_t), r_t)$ (so $\mathrm{FG}_t = \mathrm{FG}_t(f_t)$). Consider the log of the expectation of the exponent on both sides.

$$
\begin{aligned}
&\log\mathbb{E}_{f\sim p_t}\mathbb{E}_{c_t|x_t,S_t}\left[\exp\left(-\eta\left(\widehat{\ell}_{t,f} - \widehat{\ell}_{t,f^\star}\right)\right)\right] \\
&= \log\mathbb{E}_{f\sim p_t}\mathbb{E}_{c_t|x_t,S_t}\left[\exp\left(-\frac{1}{16}\widehat{\mathrm{LS}}_t + \eta\mathrm{FG}_t(f)\right)\right] \\
&\leq \frac{1}{2}\log\mathbb{E}_{f\sim p_t}\left(\mathbb{E}_{c_t|x_t,S_t}\left[\exp\left(-\frac{1}{16}\widehat{\mathrm{LS}}_t\right)\right]^2\right) + \frac{1}{2}\log\mathbb{E}_{f\sim p_t}\left[\exp\left(2\eta\mathrm{FG}_t(f)\right)\right] \\
&\leq \frac{1}{2}\log\mathbb{E}_{f\sim p_t}\left(\mathbb{E}_{c_t|x_t,S_t}\left[\exp\left(-\frac{1}{8}\widehat{\mathrm{LS}}_t\right)\right]\right) + \frac{1}{2}\log\mathbb{E}_{f\sim p_t}\left[\exp\left(2\eta\mathrm{FG}_t(f)\right)\right], \qquad (39)
\end{aligned}
$$

where the first inequality is by Cauchy-Schwarz inequality and the second inequality is because $\mathbb{E}[x]^2 \leq \mathbb{E}[x^2]$. Next, we consider bounding each of the two terms. For the first term, since

$$\left|\frac{1}{4}\sum_{i\in S_t}(\mu_i(S_t, f(x_t)) - \mu_i(S_t, f^\star(x_t)))\varepsilon_{t,i}\right| \leq \frac{\|\mu(S_t, f(x_t)) - \mu(S_t, f^\star(x_t))\|_2}{2\sqrt{2}} \leq \frac{1}{2},$$

we know that $-\frac{1}{4}(\mu(S_t, f^\star(x_t) - \mu(S_t, f(x_t)))^\top\varepsilon_t$ is a zero-mean, $\frac{1}{8}\|\mu(S_t, f(x_t)) - \mu(S_t, f^\star(x_t))\|_2^2$-sub-Gaussian random variable given $x_t$ and $S_t$, meaning that

$$\mathbb{E}_{c_t|x_t,S_t}\left[\exp\left(-\frac{1}{4}(\mu(S_t, f^\star(x_t) - \mu(S_t, f(x_t)))^\top\varepsilon_t\right)\right] \leq \exp\left(\frac{1}{16}\|\mu(S_t, f(x_t)) - \mu(S_t, f^\star(x_t))\|_2^2\right).$$

Therefore, we know that

$$\mathbb{E}_{c_t|x_t,S_t}\left[\exp\left(-\frac{1}{8}\widehat{\mathrm{LS}}_t\right)\right]$$

$$= \exp\left(-\frac{1}{8}\|\mu(S_t, f(x_t)) - \mu(S_t, f^\star(x_t))\|_2^2\right)\mathbb{E}_{c_t|x_t,S_t}\left[\exp\left(-\frac{1}{4}(\mu(S_t, f^\star(x_t)) - \mu(S_t, f(x_t)))^\top \varepsilon_t\right)\right]$$

$$\leq \exp\left(-\frac{1}{16}\|\mu(S_t, f(x_t)) - \mu(S_t, f^\star(x_t))\|_2^2\right).$$

Then, since $\frac{1}{16}\|\mu(S_t, f(x_t)) - \mu(S_t, f^\star(x_t))\|_2^2 \leq \frac{1}{8}$, using the fact that $\exp(x) \leq 1 + \frac{x}{2}$ for $x \in [-1, 0]$, we know that

$$\mathbb{E}_{c_t|x_t,S_t}\left[\exp\left(-\frac{1}{8}\widehat{\mathrm{LS}}_t\right)\right]$$

$$\leq 1 - \frac{1}{32}\|\mu(S_t, f(x_t)) - \mu(S_t, f^\star(x_t))\|_2^2$$

$$\leq 1 - \frac{1}{64(K+1)^4}\sum_{i \in S_t}(f_i(x_t) - f_i^\star(x_t))^2$$

$$= 1 - \frac{1}{64(K+1)^4}\mathrm{LS}_t,$$

where the second inequality is because Lemma B.3. Further using the fact that $\log(1 + x) \leq x$ for all $x \geq -1$, we have

$$\frac{1}{2}\log\mathbb{E}_{f\sim p_t}\left(\mathbb{E}_{c_t|x_t,S_t}\left[\exp\left(-\frac{1}{8}\widehat{\mathrm{LS}}_t\right)\right]\right) \leq -\frac{1}{128(K+1)^4}\mathrm{LS}_t. \tag{40}$$

Consider the second term in Eq. (39). Since $\eta \leq 1$ and $|\mathrm{FG}_t(f)| \leq 1$, using $e^x \leq 1 + x + 2x^2$ for $x \leq 1$, we know that

$$\frac{1}{2}\log\mathbb{E}_{f\sim q_t}\left[\exp(2\eta\mathrm{FG}_t(f))\right] \leq \frac{1}{2}\log\left(1 + 2\eta\mathbb{E}_{f\sim q_t}[\mathrm{FG}_t(f)] + 2(2\eta)^2\right)$$

$$\leq \eta\mathbb{E}_{f\sim q_t}[\mathrm{FG}_t(f)] + 4\eta^2 \qquad (\log(1+x) \leq x)$$

$$= \eta\mathbb{E}_{f_t\sim q_t}[\mathrm{FG}_t] + 4\eta^2. \qquad (f_t \text{ is drawn from } q_t)$$

Plugging the last bound and Eq. (40) into Eq. (39) and rearranging finishes the proof. $\qquad\square$

The next lemma follows the classic analysis of multiplicative weight update algorithm.

**Lemma E.3** *Algorithm 3 guarantees that for each $t \in [T]$,*

$$-\mathbb{E}\left[\mathbb{E}_{S_t\sim q_t}\log\mathbb{E}_{c_t|x_t,S_t}\mathbb{E}_{f\sim p_t}\left[\exp(-\eta(\widehat{\ell}_{t,f} - \widehat{\ell}_{t,f^*}))\right]\right] \leq Z_t - Z_{t-1},$$

*where $Z_t = -\mathbb{E}\left[\log\mathbb{E}_{f\sim p_1}\left[\exp\left(-\eta\sum_{\tau=1}^t\left(\widehat{\ell}_{t,f} - \widehat{\ell}_{t,f^*}\right)\right)\right]\right]$ and $q_t \in \Delta(\mathcal{S})$ satisfies that $q_t(S) = \mathbb{E}_{f\sim p_t}[\mathbb{1}\{S = \mathrm{argmax}_{S'\in\mathcal{S}}R(S', f(x_t), r_t)\}]$.*

**Proof** Let $G_{t,f} \triangleq \exp\left(-\eta\sum_{\tau=1}^t\left(\widehat{\ell}_{t,f} - \widehat{\ell}_{t,f^*}\right)\right)$. According to Algorithm 3, we know that

$$p_{t,f} = \frac{\exp\left(-\eta\sum_{\tau=1}^{t-1}\widehat{\ell}_{\tau,f}\right)}{\int_{f'\in\mathcal{F}}\exp\left(-\eta\sum_{\tau=1}^{t-1}\widehat{\ell}_{\tau,f'}\right)df'} = \frac{G_{t-1,f}}{\int_{f'\in\mathcal{F}}G_{t-1,f'}df'}.$$

Then, according to the definition of $Z_t$, we have

$$Z_{t-1} - Z_t$$

$$= \mathbb{E}\left[\log\frac{\int_{f\in\mathcal{F}}G_{t,f}df}{\int_{f\in\mathcal{F}}G_{t-1,f}df}\right]$$

$$= \mathbb{E}\left[\log \frac{\int_{f \in \mathcal{F}} G_{t-1,f} \exp(-\eta(\widehat{\ell}_{t,f} - \widehat{\ell}_{t,f^*}))df}{\int_{f \in \mathcal{F}} G_{t-1,f}df}\right]$$

$$= \mathbb{E}\left[\log \mathbb{E}_{f \sim p_t}\left[\exp(-\eta(\widehat{\ell}_{t,f} - \widehat{\ell}_{t,f^*}))\right]\right]$$

$$\leq \mathbb{E}\left[\mathbb{E}_{S_t \sim q_t} \log \mathbb{E}_{c_t|x_t,S_t} \mathbb{E}_{f \sim p_t}\left[\exp(-\eta(\widehat{\ell}_{t,f} - \widehat{\ell}_{t,f^*}))\right]\right],$$

where the last inequality is due to Jensen's inequality. Rearranging the terms finishes the proof. □

Next, we restate and prove Corollary 4.8.

**Corollary E.4** *Under Assumption 1, Algorithm 3 wensures* $\mathbf{Reg}_{\mathsf{MNL}} = \mathcal{O}\left(K^2 \sqrt{NT \log |\mathcal{F}|}\right)$ *for the finite class and* $\mathbf{Reg}_{\mathsf{MNL}} = \mathcal{O}\left(K^2 \sqrt{dNT \log(BTK)}\right)$ *for the linear class.*

**Proof** For a finite function class $\mathcal{F}$, since $q_1$ is uniform, we have

$$Z_T = -\mathbb{E}\left[\log \sum_{f \in \mathcal{F}} \frac{1}{|\mathcal{F}|} \exp\left(-\eta \sum_{t=1}^{T}\left(\widehat{\ell}_{t,f} - \widehat{\ell}_{t,f^*}\right)\right)\right]$$

$$\leq -\mathbb{E}\left[\log \frac{1}{|\mathcal{F}|} \exp\left(-\eta \sum_{t=1}^{T}\left(\widehat{\ell}_{t,f^*} - \widehat{\ell}_{t,f^*}\right)\right)\right] = \log |\mathcal{F}|.$$

Combining with Theorem E.1 and picking $\eta = \frac{1}{K^2}\sqrt{\frac{N \log |\mathcal{F}|}{T}}$, we prove the first conclusion.

To prove our results for the linear class, we first show a more general results for parametrized Lipschitz function class. Suppose that $\mathcal{F}$ is a $d$-dimensional parametrized function class defined as:

$$\mathcal{F} = \{f_\theta : \mathcal{X} \mapsto [0,1]^N, \|\theta\|_2 \leq B, f_{\theta,i} \text{ is } \alpha\text{-Lipschitz with respect to } \|\cdot\|_2 \text{ for all } i \in [N]\}. \tag{41}$$

Direct calculation shows that the linear function class we consider is an instance of Eq. (41) with $\alpha = 1$. For function class satisfying Eq. (41), we aim to show that $Z_T = \mathcal{O}(K\eta + d \log(\alpha BT))$. Specifically, we consider a small $\ell_2$-ball around the true parameter $\theta^*$: $\Omega_T = \{\theta : \|\theta - \theta^*\|_2 \leq \frac{1}{\alpha TK}\}$. Since $\mathcal{F}$ is $\alpha$-Lipschitz with respect to $\|\cdot\|_2$, we know that for any $x \in \mathcal{X}$, and any $i \in [N]$,

$$|f_{\theta,i}(x) - f_{\theta^*,i}(x)| \leq \frac{1}{TK}. \tag{42}$$

Therefore, for any $\theta \in \Omega_T$,

$$-\eta(\widehat{\ell}_{t,f_\theta} - \widehat{\ell}_{t,f_{\theta^*}})$$

$$= -\frac{1}{16}\sum_{i \in S_t}(f_{\theta,i}(x_t) - c_{t,i})^2 + \frac{1}{16}\sum_{i \in S_t}(f_{\theta,i}(x_t) - c_{t,i})^2$$

$$\quad + \eta \cdot \max_{S \in \mathcal{S}} R(S, f_\theta(x_t), r_t) - \eta \cdot \max_{S \in \mathcal{S}} R(S, f_{\theta^*}(x_t), r_t)$$

$$\geq -\frac{1}{8}\sum_{i \in S_t}|f_{\theta,i}(x_t) - f_{\theta^*,i}(x_t)| + \eta \cdot \max_{S \in \mathcal{S}} R(S, f_\theta(x_t), r_t) - \eta \cdot \max_{S \in \mathcal{S}} R(S, f_{\theta^*}(x_t), r_t). \tag{43}$$

Let $S(f_{\theta^*}(x_t), r_t) = \operatorname{argmax}_{S \in \mathcal{S}} R(S, f_{\theta^*}(x_t), r_t)$. Then, we can further lower bound Eq. (43) as follows:

$$-\eta(\widehat{\ell}_{t,f_\theta} - \widehat{\ell}_{t,f_{\theta^*}})$$

$$\geq -\frac{1}{8}\sum_{i \in S_t}|f_{\theta,i}(x_t) - f_{\theta^*,i}(x_t)| + \eta R(S(f_{\theta^*}(x_t), r_t), f_\theta(x_t), r_t) - \eta \cdot \max_{S \in \mathcal{S}} R(S, f_{\theta^*}(x_t), r_t)$$

$$\overset{(i)}{\geq} -\frac{1}{8}\sum_{i \in S_t}|f_{\theta,i}(x_t) - f_{\theta^*,i}(x_t)| - \eta \sum_{i \in S(f_{\theta^*}(x_t), r_t)}|f_{\theta,i}(x_t) - f_{\theta^*,i}(x_t)|$$

$$\overset{(ii)}{\geq} -\frac{1}{8T} - \frac{\eta}{T},$$

where $(i)$ is because Lemma B.1 and $(ii)$ uses Eq. (42). This means that

$$
\begin{aligned}
Z_T &= -\mathbb{E}\left[\log \mathbb{E}_{f \sim q_1} \exp\left(-\eta \sum_{t=1}^{T}\left(\widehat{\ell}_{t,f_\theta} - \widehat{\ell}_{t,f_{\theta^\star}}\right)\right)\right] \\
&\leq -\mathbb{E}\left[\log(\alpha BT)^{-d} \inf_{\theta \in \Omega_T} \exp\left(-\eta \sum_{t=1}^{T}\left(\widehat{\ell}_{t,f_\theta} - \widehat{\ell}_{t,f_{\theta^\star}}\right)\right)\right] \\
&\leq d\log(\alpha BT) + \frac{1}{8} + \eta = \mathcal{O}(K\eta + d\log(\alpha BKT)).
\end{aligned}
$$

With the optimal choice of $\eta = \frac{1}{K^2}\sqrt{\frac{Nd\log(\alpha BTK)}{T}}$, Theorem E.1 shows that Algorithm 3 guarantees that for linear function class

$$
\mathbf{Reg}_{\mathsf{MNL}} = \mathcal{O}\left(K^2\sqrt{dNT\log(BTK)}\right).
$$

$\square$

