# OpenReview forum: "Contextual Multinomial Logit Bandits with General Value Functions"
_NeurIPS.cc/2024/Conference — NeurIPS 2024 poster_

### Official Review · Reviewer_AgtJ · 2024-06-16

**Soundness:** 3
**Presentation:** 3
**Contribution:** 3
**Rating:** 5
**Confidence:** 2

**Summary:**

This paper considers MNL bandits with a general value function. The authors first examine the case of stochastic contexts and rewards. They suggest an epoch-based algorithm with an offline regression oracle. With uniform exploration, the algorithm achieves a regret bound, specifically $T^{2/3}$ for finite and linear classes. By utilizing better exploration, it achieves $\sqrt{T}$ for these classes. Next, they consider adversarial contexts and rewards. With uniform exploration, the algorithm achieves a regret bound of $T^{5/6}$ for finite or linear classes. With better exploration, it achieves $T^{3/4}$. Lastly, by using Thompson sampling, it achieves a $\sqrt{T}$ regret bound. Importantly, the suggested algorithms do not depend on $\kappa$, implying better dependency on $B$.

**Strengths:**

- This paper first considers a general value function for MNL bandits.

- They propose algorithms for stochastic or adversarial context and rewards and provide regret analysis.

- The regret bound has better dependency on $B$ (without including $\kappa$) compared to previously suggested ones.

**Weaknesses:**

1. The regret bound has supper linear dependency on $K$ for $\sqrt{T}$ in Corollayr 3.8, 4.7, 4.8.
2. It does not provide regret lower bounds so it is hard to know the tightness of the achieved regret upper bounds.

**Questions:**

1. Is there any insight about how regret bounds do not include the dependency on $\kappa$ for linear class? Or does it include $\kappa$ when it is applied to the standard contextual MNL model?

2. Feel-good Thompson sampling algorithm seems to outperform other algorithms including stochastic or adversarial cases. Is there a benefit to using epoch-based algorithms over the Thompsom sampling method?

**Limitations:**

I could not find a discussion on the limitations of this work.

---

> ### Author Rebuttal · Authors · 2024-08-06
>
> **Q1: Is there any insight about how regret bounds do not include the dependency on 𝜅 for linear class? Or does it include 𝜅 when it is applied to the standard contextual MNL model?**
>
> A: The intuition on why we do not have $\kappa$ dependency is that most previous MNL works (e.g. [7,20,23]) adopt a UCB-type approach, which first approximates the true weight parameter $\theta$ via MLE and constructs a confidences set based on this approximation. The $\kappa$ dependence will unavoidably appear from this approximation step. Different from previous approaches, we reduce the learning problem to a regression problem, and show that the regret directly depends on the regression error. This key step enables us to remove the dependency on $\kappa$.
>
> **Q2: Feel-good Thompson sampling algorithm seems to outperform other algorithms including stochastic or adversarial cases. Is there a benefit to using epoch-based algorithms over the Thompson sampling method?**
>
> A: As shown in Table 1, while Feel-good Thompson sampling outperforms other algorithms in terms of regret upper bound, it is not computationally efficient (even for a small $K$) since it is unclear how to sample the value function at each round efficiently. On the other hand, our other algorithms, including those epoch-based algorithms, are computationally efficient (at least for small $K$).
>
> **Q3: I could not find a discussion on the limitations of this work.**
>
> A: We do discuss the limitations of our work in various places (as acknowledged by other reviewers), such as the assumption on the no-purchase option, the inefficiency of solving Eq. (5) when K is large (Lines 214-215), and the disadvantage of FGTS algorithm (Lines 322-326),

---

> ### Comment · Reviewer_AgtJ · 2024-08-08
> **Thank you for your response**
>
> For the $\kappa$, it is still not clear how to avoid this term. For the regression with linear class, ERM seems to minimize log-loss to estimate parameters as in the previous MNL model, which includes approximate parameter. Could you provide any helpful comments regarding this? I'm also wondering about that the $\epsilon$-covering number does not need to include $\kappa$.

---

> > ### Author Response · Authors · 2024-08-08
> >
> > To clarify, the reason why most previous MNL works using UCB-type approach (e.g. [7,20,23]) have this $\kappa$ dependency is that their analysis depends on the confidence width constructed via MLE (e.g.Theorem 2 in [20]). However, different from previous approaches, we show that the regret directly depends on the **regression error**. Moreover, while the ERM oracle may involve parameter approximation, importantly its regression error **does not** explicitly depends on the distance between the estimation and the true parameter, because making accurate predictions is in a sense easier than making accurate estimation of the parameter (an inaccurate estimation on the parameter could still lead to an accurate prediction).
> >
> >
> > As for the covering number, we show in Appendix B.1 (line 427) that the $\epsilon$-covering number is bounded by $\left(\frac{16B}{\epsilon}\right)^d$, with no dependence on $\kappa$.

---

> > > ### Comment · Reviewer_AgtJ · 2024-08-10
> > > **Thank you for your response.**
> > >
> > > I appreciate your detailed explanation addressing the concern regarding $\kappa$. However, the regret bound has a superlinear dependency on $K$, which reduces the benefit of removing $\kappa$ when the norm of the parameter is bounded by 1. Given my primary concern about the tightness of the bound, I am maintaining my score.

---

### Official Review · Reviewer_a6sS · 2024-07-09

**Soundness:** 3
**Presentation:** 3
**Contribution:** 3
**Rating:** 6
**Confidence:** 3

**Summary:**

This paper addresses the problem of contextual multinomial logit (MNL) bandits with general value functions across both stochastic and adversarial settings. The authors develop a suite of algorithms for different settings and with different computation-regret trade-offs. The application to the linear case surpasses previous works in terms of both statistical and computational efficiency.

**Strengths:**

1. **Novelty of the Setting**: This paper is the first to explore contextual MNL bandits with general value functions, representing a significant expansion in the scope of MNL bandit problems. The setting is both novel and interesting.
2. **Innovative Techniques**: The introduction of several new techniques to tackle the complexities introduced by general value functions is commendable. The methods may inspire the following works and be useful in other areas.
3. **Improved Efficiency**: The application of these methods to linear cases shows improvements over previous works in both statistical and computational efficiency, making this a valuable contribution to the field.

**Weaknesses:**

1. **Computational Inefficiency**: The Feel-Good Thompson sampling algorithm, as discussed, lacks computational efficiency, even for linear cases, which could limit its practical applicability.
2. **Lack of Experimental Validation**: The absence of empirical experiments to verify the theoretical claims weakens the paper's impact. Experimental results are crucial for validating the effectiveness and practicality of the proposed methods.

**Questions:**

1. Zhang and Sugiyama (2023) developed a computationally efficient algorithm for MLogB bandit problem. As MLogB and MNL are similar, how might their approach be adapted to the MNL bandit problem addressed in this paper to enhance computational efficiency?
2. The authors claim that to ensure that no regret is possible, they make Assumption 1 in Line 96. Does this imply that achieving no regret is impossible in unrealizable scenarios? Could the authors provide some intuition about the reason?

Ref: Yu-Jie Zhang and Masashi Sugiyama. Online (Multinomial) Logistic Bandit: Improved Regret and Constant Computation Cost. In NeurIPS 2023.

---

> ### Author Rebuttal · Authors · 2024-08-06
>
> **Q1: Computational Inefficiency: The Feel-Good Thompson sampling algorithm, as discussed, lacks computational efficiency, even for linear cases, which could limit its practical applicability.**
>
> A: We acknowledge that the Feel-Good Thompson sampling is not efficient theoretically. However, empirically, as mentioned in [30], one can apply stochastic gradient Langevin dynamic (SGLD) to approximately sample a value function.
>
> **Q2: [Zhang and Sugiyama, 2023] developed a computationally efficient algorithm for MLogB bandit problem. As MLogB and MNL are similar, how might their approach be adapted to the MNL bandit problem addressed in this paper to enhance computational efficiency?**
>
> A: While MLogB and MNL share some common components, their setups are in fact quite different: MNL considers the case where multiple items (decision) are chosen at each round and one of these items is selected according a multinomial distribution, while MLogB considers the case where a single item (decision) is chosen at each round but the outcome is one of the $K+1$ different possible outcomes (including the no-purchase outcome) following a multinomial distribution. Since the computational inefficiency for MNL usually comes from the fact that the number of possible subsets is large, we do not see how ideas from MLogB can be utilized to improve the computational efficiency of our algorithms.
>
> **Q3: The authors claim that to ensure that no regret is possible, they make Assumption 1 in Line 96. Does this imply that achieving no regret is impossible in unrealizable scenarios? Could the authors provide some intuition about the reason?**
>
> A: Making a realizability assumption is standard for bandit problems with function approximation [10, 11, 14, 29, 31], usually for the purpose of obtaining efficient algorithms via reduction to regression, but it does not imply that no regret is impossible without it. In fact, for contextual bandits, applying the classical (but inefficient) Exp4 algorithm [Auer et al., 2002] achieves the optimal regret even without the realizability assumption.
>
> [Auer et al., 2002]: Peter Auer, Nicolò Cesa-Bianchi, Yoav Freund, and Robert E. Schapire, The Nonstochastic Multiarmed Bandit Problem, SIAM Journal on Computing, 2002.

---

> > ### Comment · Reviewer_a6sS · 2024-08-09
> > **Thank you for your rsponse.**
> >
> > I thank the authors for their response and have no further questions.

---

### Official Review · Reviewer_8kDH · 2024-07-09

**Soundness:** 2
**Presentation:** 2
**Contribution:** 3
**Rating:** 4
**Confidence:** 5

**Summary:**

This paper introduces a couple of algorithms for contextual multinomial logit bandits under two different assumptions: i) stochastic contexts and rewards; ii) adversarial contexts and rewards. The theoretical analysis for algorithms for these two setups is pretty solid. Despite the contribution of this study, I believe that this paper needs more work to be done for acceptance.

**Strengths:**

The setups for this work are pretty inclusive, representing that the contributions of the work can be significant. The literature review is also solid as well.

**Weaknesses:**

The paper seems incomplete, possibly due to page limits. Some algorithms and results are not fully described, and there is a lack of experimental validation to support the theoretical findings. The authors should better organize and present their work, focusing on the most important results. Additionally, many terms and mathematical notations are used without proper definitions or introductions.

In spite of repeated appearances of log loss regression, its definition has not been stated.
The definitions of Err_log, Reg_log, and ERM are missing.
For the function class F, what is the definition of |F| in Lemma 3.1?
I think that many things other than these are missing.

**Questions:**

What is the relationship between \pi and q_m?

I am interested in how to apply the feel-good TS [30] for this setup.

**Limitations:**

Thank the authors for mentioning some limitations in the manuscript. The reviewer also agreed that solving eq (5) with polynomial time complexity is not easy work.

---

> ### Author Rebuttal · Authors · 2024-08-06
>
> **Q1: The paper seems incomplete, possibly due to page limits. Some algorithms and results are not fully described. Additionally, many terms and mathematical notations are used without proper definitions or introductions.**
>
> A: We strongly disagree with this comment. Could the reviewer kindly point out what algorithms and results are not fully described in this paper? For your examples of “terms and mathematical notations used without proper definitions or introductions”, they are in fact all explicitly defined (see below).
>
> **Q2: In spite of repeated appearances of log loss regression, its definition has not been stated.**
>
> A: This is not true. The definition of log loss regression is in Assumption 2 (offline) and Assumption 3 (online), with the definition of log loss in line 77.
>
>
> **Q3: The definitions of Err_log, Reg_log, and ERM are missing.**
>
> A: $\text{Err}_{\log}$ is defined in Assumption 2 and meant to be an abstract generalization error bound of the offline regression oracle, whose concrete form depends on its three arguments:
> $n$ (the number of samples), $\delta$ (the failure probability), and $\mathcal{F}$ (the function class) and is instantiated clearly in Lemma 3.1 and Lemma D.1 for three examples.
>
> Similarly, $\text{Reg}_{\log}$ is defined in Assumption 3 and meant to be an abstract regret bound of the online regression oracle. Again, its concrete form for three examples are provided in Lemma 4.1 and Lemma D.1.
>
> We also emphasize that deriving regret bounds with a general generalization error or regret bound of the regression oracle and then instantiating them for concrete examples is very common in this line of work; see [10, 11, 14, 25, 27, 29, 31].
>
> ERM, “empirical risk minimizer” (line 119), is explicitly defined in Lemma 3.1 (line 123).
>
> **Q4: For the function class $\mathcal{F}$, what is the definition of $|\mathcal{F}|$ in Lemma 3.1?**
>
> A: $|\mathcal{F}|$ represents the cardinality of the finite set $\mathcal{F}$ (which is a rather standard notation).
>
> **Q5: What is the relationship between $\pi$ and $q_m$?**
>
> A: $q_m: (\mathcal{X}\times r) \mapsto \Delta(\mathcal{S})$ is a stochastic policy (mapping from a context-reward pair to a distribution over subsets (line 136)) that Algorithm 1 decides in each epoch m, while $\pi: (\mathcal{X}\times r) \mapsto \mathcal{S}$ is a deterministic policy (mapping from a context-reward pair to a subset (line 143)) that appears in our analysis. We are not very certain about what specific “relationship” the reviewer is asking about.
>
> **Q6: I am interested in how to apply the feel-good TS [30] for this setup.**
>
> A: All details on applying feel-good TS to our setup can be found in Appendix E (as mentioned in Sec 4.2).

---

> > ### Comment · Reviewer_8kDH · 2024-08-14
> >
> > I thank the authors for their detailed response. I will make some changes in my evaluation based on this.

---

### Official Review · Reviewer_z4QH · 2024-07-11

**Soundness:** 3
**Presentation:** 3
**Contribution:** 3
**Rating:** 6
**Confidence:** 3

**Summary:**

The paper presents three primary contributions for the contextual multinomial logit bandits considering both stochastic and adversarial contexts and rewards.

-  a suite of algorithms proposed each with a different computation-regret trade-off.

- advances existing regrets by removing the dependence on certain problem-dependent constants.

- extends existing works to study a general value function class (1-Lipschitz function class).

**Strengths:**

- This seems to be the first contextual multinomial logit bandit paper considering adversarial context and reward with online regression oracle. I think the community will find this interesting.

- The paper is generally well-written and technically solid.

**Weaknesses:**

- No experiments and simulation results are provided. But to be fair, there are other theoretical papers without experimental illustrations.

- For stochastic contexts and adversarial rewards, [20] show there exists an efficient algorithm achieving $O(\sqrt{T})$ regret. The adversarial reward setup is more general than the stochastic reward. So it is fair to compare it to corollary 3.5. Algorithm 1 has a larger regret upper bound $O( T^{\frac{2}{3}})$.

**Questions:**

- Is this the first MNL bandit paper considering adversarial context and reward with online regression oracle?

- Could you please address the second point in Weakness?

- It is understandable that Algorithm 1 uses a doubling trick. But removing sampling history when feeding to offline regression oracle (4th line in Algorihtm 1) is not sensible in my opinion. Could you address this issue without affecting much of the analysis?

- Since the paper studied MNL bandit with a general value function class (1-Lipschitz function class), do you think the regret guarantee can be characterized via eluder dimension?

The next two questions are related.

- It is claimed after Corollary 3.5 that Algorithm 1 achieves a smaller regret dependence on $B$. Why is $B$ a parameter related to regret or $\textbf{Err}_{\log}$? According to the definition of linear class in Lemma 3.1, it is equivalent to assigning weight $e^{\theta^\top x_i}$ to item $i$ and $e^B$ to no-purchase. Since it is required that $\lVert x_i \rVert_2 \leq 1$ and $\lVert \theta \rVert_2 \leq B$, it seems that all $B > 0$ are equivalent.

- It is understandable the paper follows  [5, 9, 18] to assume the no-purchase option is the most likely outcome. Is this a necessary assumption without which the algorithm could fail? Is it possible to relax the assumption to something like "the probability (or the weight in equation 1) of no-purchase is lower bounded by some $\Delta>0$"? I could imagine that the regrets are related to $\Delta$ (other than $B$ ) since it can be hard to learn $f$ with a large probability of the no-purchase option.

Some minor issues:

- In line 173, should "$\epsilon \rightarrow \epsilon_m$"?

- The optimal choice of $\epsilon_m$ and $\gamma_m$ are given in Appendix. I suggest including them in the Algorithm sections to make the main paper self-contained.

- If extending contextual MNL bandits to a general value function class setup is a major contribution, I would suggest to move the paragraph into the main paper.

**Limitations:**

The authors adequately addressed the limitations.

---

> ### Author Rebuttal · Authors · 2024-08-06
>
> **Q1: For stochastic contexts and adversarial rewards, [20] show there exists an efficient algorithm achieving $𝑂(\sqrt {𝑇})$ regret. The adversarial reward setup is more general than the stochastic reward. So it is fair to compare it to corollary 3.5. Algorithm 1 has a larger regret upper bound $O(T^{2/3})$.**
>
> A: Note that we did compare to [20] in the discussion after Corollary 3.5 (Line 195 more specifically) and highlighted that the regret bound of [20] depends on $\kappa$ which can be exponential in $B$ in the worst case, while our bound only has polynomial dependence on $B$. We will add that our bound indeed has a worse dependence on $T$ as suggested.
>
> **Q2: Is this the first MNL bandit paper considering adversarial context and reward with online regression oracle?**
>
> A: Yes, to the best of our knowledge, our work is the first one considering the case where both the context and the reward can be adversarial (with or without regression oracles).
>
> **Q3: It is understandable that Algorithm 1 uses a doubling trick. But removing sampling history when feeding to offline regression oracle (4th line in Algorithm 1) is not sensible in my opinion. Could you address this issue without affecting much of the analysis?**
>
> A: While it might seem sensible to feed all the sampling history to the offline regression oracle due to the stochasticity of contexts and rewards, it does not work since the oracle requires **i.i.d. inputs of context-subset-purchase tuples** (not i.i.d. context-reward), and different epochs of our algorithm create different distributions over these tuples.
>
> **Q4: Since the paper studied MNL bandit with a general value function class (1-Lipschitz function class), do you think the regret guarantee can be characterized via eluder dimension?**
>
> A: Our regret bounds explicitly depend on the generalization error of ERM oracle in the stochastic environment, and the online regression regret (Algorithm 2) or the log partition function (Algorithm 3) in the adversarial environment. It is unclear to us how these quantities are directly related to the eluder dimension of a function class.
>
> **Q5: It is claimed after Corollary 3.5 that Algorithm 1 achieves a smaller regret dependence on $𝐵$. Why is $𝐵$ a parameter related to regret or $\text{Err}_{\log}$? According to the definition of linear class in Lemma 3.1, it is equivalent to assigning weight to item 𝑖 and $𝑒^𝐵$ to no-purchase. Since it is required that $||x_i||_2\leq 1$ and $\|\|\theta\|\|_2\leq B$, it seems that all $𝐵>0$  are equivalent.**
>
> A: Your reparameterization is correct. However, a larger parameter $B$ makes the lowest possible weight for each item, $e^{-B}$ under your parameterization, smaller, which in turns makes the scale of the log loss larger and naturally increases the generalization error $\text{Err}_{\log}$.
>
> **Q6: It is understandable the paper follows [5, 9, 18] to assume the no-purchase option is the most likely outcome. Is this a necessary assumption without which the algorithm could fail? Is it possible to relax the assumption to something like "the probability (or the weight in equation 1) of no-purchase is lower bounded by some $\Delta>0$"? I could imagine that the regrets are related to $\Delta$ (other than $𝐵$) since it can be hard to learn $𝑓$ with a large probability of the no-purchase option.**
>
> A: If the weight for no-purchase is lower bounded by $\Delta\in(0,1)$ instead of $1$, all our regret bounds still hold with an extra factor of $1/\Delta$ (explained below). Note that this is the opposite to what the reviewer suggested — while a large no-purchase probability seemingly makes it harder to learn $f^\star$ at first glance, it in fact makes it easier because it leads to a better reverse Lipschitzness of $\mu$, and consequently the difference in value is better controlled by the difference in log loss. As an extreme example, suppose that the no-purchase weight is 0. Then, no matter what value an item has, it will always be purchased by the customer if the selected subset contains only this item, revealing no information to the learner at all.
>
> To see why our bounds hold with an extra factor of $1/\Delta$, note that the only modification needed is Lemma B.3 (reverse Lipschitz Lemma): in this case we need to show the reverse Lipschitzness for $h(a) = a/(\Delta+1^\top a)$, and via a similar calculation, it can be shown that $\Omega(\Delta^2/d^4)\|\|a-b\|\|^2 \leq \|\|h(a)-h(b)\|\|^2.$ The remaining analysis remains the same.
>
> Thanks for pointing this out; we will add this discussion to the next version.
>
>
> Thanks for the other suggestions and pointing out the typos. We will incorporate these in the next revision.

---

> > ### Comment · Reviewer_z4QH · 2024-08-12
> > **Response to rebuttal**
> >
> > We thank the authors for their response. I am maintaining my score since it is the first MNL bandit paper considering adversarial context and reward with online regression oracle. However, I hope the authors could address the following issues or give better explanation.
> >
> > - Q3: A iid requirement is incorrect. There is no identical distribution. I gues the author means conditional independent given the recommendation actions. So the entire history can be used in my opinion.
> >
> > - Q5: I still believe $B$ should not be included in the regret upper bound and used in comparison with other algorithm. It is a free parameter and reparameterization does not change the problem definition. A larger $B$ will not affect log loss since $\theta$ will also scale up. If this is right, the comparison with [20] should either be removed or revised.
> >
> > - Q6: This is just a suggestion. The paper could benefit from introducing our discussion on $\Delta$ into the problem definition and analysis. An intuitive explaination on how it affects regret could give readers a better understanding of the MNL bandit problem.

---

> > > ### Author Response · Authors · 2024-08-13
> > >
> > > We thank the reviewer for the further discussions and we address your questions as follows.
> > >
> > > **Q3: A iid requirement is incorrect. There is no identical distribution. I guess the author means conditionally independent given the recommendation actions. So the entire history can be used in my opinion.**
> > >
> > > A: No, we indeed mean that the tuples $(x, S, i)$ collected within epoch $m$ are i.i.d. drawn from the same distribution. This is true since the selection of $S$ follows a **fixed** conditional distribution $q_m(x,r)$ given the context $x$ and the reward $r$ within epoch $m$ (and $(x,r)$ are i.i.d. of course).
> > >
> > > On the other hand, this i.i.d. property does not hold for tuples coming from different epochs since $q_m(x,r)$ is different for different epochs. This is why we cannot use the entire history.
> > >
> > > **Q5: I still believe 𝐵 should not be included in the regret upper bound and used in comparison with other algorithm. It is a free parameter and reparameterization does not change the problem definition. A larger 𝐵 will not affect log loss since 𝜃 will also scale up. If this is right, the comparison with [20] should either be removed or revised.**
> > >
> > > A: Note that we already normalize the contexts so that they are within the unit $\ell_2$ ball, so further normalizing $\theta$ is not without loss of generality and is restricting the representation power of the function class.
> > >
> > > We also do not understand the comment “A larger 𝐵 will not affect log loss since 𝜃 will also scale up”. Specifically, a larger $B$ in our formulation can lead to a lower probability of item $i$ being selected since the value of this item can be as low as $e^{-2B}$, which eventually makes the scale of the log loss to be of order $B$ as shown between Line 425 and Line 426.
> > >
> > > **Q6: This is just a suggestion. The paper could benefit from introducing our discussion on Δ into the problem definition and analysis. An intuitive explanation of how it affects regret could give readers a better understanding of the MNL bandit problem.**
> > >
> > > A: Thanks for your suggestion. We will incorporate this into our next revision.

---

> ### Comment · Reviewer_z4QH · 2024-08-13
> **Response**
>
> Thanks to the authors for their response and for making these points clear. Now it is clear how $B$ affects regret and what iid means here. I hope the authors can explain the following questions in the revised paper. At least they are not very intuitive to me.
>
> - Why IID is necessary for the algorithm? We can still compute $Err_{\log}$ given $S$. It is known IID assumption with fixed $q_m$ is not a requirement for Thompson sampling. So it is not very intuitive to me. What aspects of the problem make it necessary here?
>
> - The relationship between $B$ and $\Delta$ still gives me contradictory information. Bigger $B$ and $\Delta$ result in a lower probability of item $i$ being selected. Why does the regret increase with $B$ but decrease with $\Delta$?
>
> I hope these questions are helpful. Again, I'll maintain my rating on this paper.

---

> ### Author Response · Authors · 2024-08-13
>
> We thank the reviewer for the questions again. They are indeed very helpful, and we will add more explanation to the paper. Specifically:
>
> **Why IID is necessary for the algorithm? We can still compute $\text{Err}_{\log}$ given $S$. It is known IID assumption with fixed $q_m$ is not a requirement for Thompson sampling. So it is not very intuitive to me. What aspects of the problem make it necessary here?**
>
> - It is necessary to have i.i.d. data in our algorithm design solely because we assume that our oracle requires i.i.d. data as inputs. It is completely possible that to solve the problem itself, using the entire history is a feasible solution, but note that the weaker the oracle assumption we make, the stronger our result is, so we choose to assume that the oracle only works with i.i.d. data.
>
> **The relationship between $\Delta$ and $B$ still gives me contradictory information. Bigger $B$ and $\Delta$ result in a lower probability of item $i$ being selected. Why does the regret increase with $B$ but decrease with $\Delta$?**
>
> - Technically, $\Delta$ and $B$ have opposite effects because a larger $\Delta$ makes it easier for the learner to identify the value difference from the realized item selection, while a larger $B$ makes the regression on the item selection probability harder. This is a great question though, and we will add more intuitive explanations to the paper.

---

### Decision · Program_Chairs · 2024-09-25

**Decision:**

Accept (poster)

**Comment:**

The paper is the first to address the contextual multinomial logit bandit problem with general value functions. New algorithms are proposed under two settings, the stochastic iid setting and the adversarial setting. Algorithms using regression oracles and the log-barrier regularized strategy are proposed, which are the first that do not depend on problem-specific constant that can be exponentially large in the norm of weights.

Reviewers acknowledge the innovativeness of the proposed method and the contribution to the more challenging setting (with general value function class) than the setting addressed before (with generalined linear function class).

Authors are encouraged to incorporate the following comments in their final version of the paper:

- super-linear dependency in K : while the proposed method removes dependency on $\kappa$ in the regret bound, it has super-linear dependency on K. This limitation should be mentioned in the text.
- reason why iid assumption is needed should be stated in detail
- reason why dependence on B is unavoidable should be stated in detail (ex, as the authors stated in their rebuttal, further normalizing theta would restrict the representation power of the function class)